# Pangenome-wide association study reveals the selective absence of CRISPR genes (Rv2816c-19c) in drug-resistant *Mycobacterium tuberculosis*

Nikhil Bhalla,[1] Ranjan Kumar Nanda[1]

**ABSTRACT** The presence of intermittently dispersed insertion sequences and transposases in the *Mycobacterium tuberculosis* (Mtb) genome makes intra-genome recombination events inevitable. Understanding their effect on the gene repertoires (GR), which may contribute to the development of drug-resistant Mtb, is critical. In this study, publicly available WGS data of clinical Mtb isolates (endemic region $n$ = 2,601; non-endemic region $n$ = 1,130) were *de novo* assembled, filtered, scaffolded into assemblies, and functionally annotated. Out of 2,601 Mtb WGS data sets from endemic regions, 2,184 (drug resistant/sensitive: 1,386/798) qualified as high quality. We identified 3,784 core genes, 123 softcore genes, 224 shell genes, and 762 cloud genes in the pangenome of Mtb clinical isolates from endemic regions. Sets of 33 and 39 genes showed positive and negative associations ($P$ < 0.01) with drug resistance status, respectively. Gene ontology clustering showed compromised immunity to phages and impaired DNA repair in drug-resistant Mtb clinical isolates compared to the sensitive ones. Multidrug efflux pump repressor genes (Rv3830c and Rv3855c) and CRISPR genes (Rv2816c-19c) were absent in the drug-resistant Mtb. A separate WGS data analysis of drug-resistant Mtb clinical isolates from the Netherlands ($n$ = 1130) also showed the absence of CRISPR genes (Rv2816c-17c). This study highlights the role of CRISPR genes in drug resistance development in Mtb clinical isolates and helps in understanding its evolutionary trajectory and as useful targets for diagnostics development.

**IMPORTANCE** The results from the present Pan-GWAS study comparing gene sets in drug-resistant and drug-sensitive Mtb clinical isolates revealed intricate presence-absence patterns of genes encoding DNA-binding proteins having gene regulatory as well as DNA modification and DNA repair roles. Apart from the genes with known functions, some uncharacterized and hypothetical genes that seem to have a potential role in drug resistance development in Mtb were identified. We have been able to extrapolate many findings of the present study with the existing literature on the molecular aspects of drug-resistant Mtb, further strengthening the relevance of the results presented in this study.

**KEYWORDS** pangenome, multidrug resistance, antimicrobial resistant evolution, tuberculosis, GWAS, DR-determinants, gene repertoires

A recent report by the World Health Organization (WHO) indicated 410,000 new drug-resistant Tuberculosis (TB) cases in 2022. TB has a higher prevalence in tropical regions, such as South Asia and Africa, that mainly comprise poverty-stricken developing nations, compared to other parts of the world. The causative agent of TB, that is, *Mycobacterium tuberculosis* (Mtb), is a slow-growing pathogen that mostly causes lower respiratory tract infections. Mtb has several intrinsic drug resistance-conferring factors

**Ad Hoc Peer Reviewers** Sudha Ramaiah, Vellore Institute of Technology, Vellore, India; Nena A., Azeezia Institute of Medical Sciences & Research, Kollam, Kerala, India; Kaan Çeylan, Faculty of Medicine University of Gaziantep, Gaziantep, Turkey

Address correspondence to Nikhil Bhalla, nikhilbhalla94@gmail.com, or Ranjan Kumar Nanda, ranjan@icgeb.res.in.

The authors declare no conflict of interest.

See the funding table on p. 13.

including thick cell walls, lipid-rich cell membranes, drug-inactivating enzymes, and drug target modification systems. Host-dependent selective pressures such as incompliance, inappropriate antibiotic selection, and dosage, and factors like delayed diagnosis lead to the accumulation of mutations in specific Mtb genes that result in ineffective drug action (1–4). Despite the introduction of several novel and repurposed drugs for the treatment of drug-resistant TB, unresponsiveness to new therapeutics is increasingly being reported (5). Mtb's slow growth rate, ability to remain dormant for decades, development of drug tolerance, and drug resistance also contribute to the relapse of TB, which is a growing concern for TB elimination programs and needs a better understanding of the gene repertoire (GR) levels.

Drug resistance in Mtb is primarily caused by the acquisition of single nucleotide polymorphisms (SNPs) in the drug target genes, prodrug-activating genes, their promoter regions, and other genes that are involved in the mechanism of action of respective drugs. Unlike in other bacteria, Horizontal Gene Transfer (HGT) of the drug resistance-conferring plasmids does not contribute to the development of drug resistance in Mtb (6–8). However, Mtb is susceptible to infection by *Mycobacteriophages* and can also undergo natural intra-genome recombination events (9, 10). In a recent study, the insertion sequence IS6110 that encodes a transposase is found to actively take part in transposition, thereby leading to genetic deletions in an observable time frame of 1 year (11). IS6110 is also reportedly more abundant in Lineage 2 (Beijing) Mtb isolates, which are widely known to be highly drug resistant (11, 12). Events like gene deletions can directly contribute to the emergence of drug resistance or indirectly by improving fitness.

Genome-wide association studies (GWAS) have been extensively used to identify drug resistance-associated SNPs in Mtb, but its application at the pangenome level remains limited (13). Existing literature on Pan-GWAS of Mtb primarily focused on identifying the genetic determinants taking part in higher prevalence, the site of infection (extrapulmonary/pulmonary), and those forming inter-species diversity (14–16). A Pan-GWAS study from 2018 identified 24 novel genetic signatures associated with drug resistance using WGS data of 1,595 clinical Mtb isolates from varying geographical regions (17). Another report was more focused on understanding the causation of atypical drug resistance in Mtb isolates (18). In this, several unique genes, as well as intergenic regions, are found to be exclusively associated with atypical drug resistance in Mtb compared to 145 other isolates with typical drug-resistant markers.

Given the limited literature on the pangenome of drug-resistant Mtb isolates and our understanding of genomic structural variations that may arise upon drug resistance development, we aimed to identify unique GRs by analyzing publicly available Mtb WGS data sets from TB endemic and non-endemic regions. The identified gene sets in the present study might be useful in understanding the evolution of drug resistance development in Mtb, identifying novel drug targets, and developing molecular tests for diagnostics development.

## MATERIALS AND METHODS

### Data acquisition, *De novo* assembly, and metagenome detection

From existing reports with corresponding NCBI-BioProjects: PRJNA879962, PRJEB41116, PRJEB32684, PRJNA379070, and PRJEB29435, Mtb WGS data ($n$ = 2,601) from TB-endemic regions (India, China, Pakistan, and Zambia) were downloaded from the NCBI-Sequence Read Archive (SRA) using fasterq-dump v2.11.3 of the SRA toolkit (NCBI) (19–23). For comparative analysis, the Mtb WGS data (PRJEB32037, $n$ = 1,130) from a TB non-endemic country, that is, the Netherlands were downloaded from the European Nucleotide Archive (ENA) in BAM format (24). Samtools bam2fq option was used for converting the BAM files to interleaved FASTQ format. Similar data processing methods were employed for all Mtb WGS data. Megahit assembler v1.2.9. was used for the *de novo* assembly of Mtb WGS data (25). Ragtag v2.1.0. was used for the scaffolding of Megahit assemblies

(26). Kraken2 v2.1.3. was used for identifying metagenomes using Minikraken database (https://ccb.jhu.edu/software/kraken/dl/minikraken_20171019_8GB.tgz) (27). The quality of *de novo* assemblies was determined using QUAST v5.2.0 (28).

## Drug resistance profiling and gene repertoire analysis

For drug resistance profiling and lineage identification of the Mtb isolates, the Megahit assemblies were used as input for TBprofiler v5.0.0. (Database: n0599ccdEJody) and the output data were compiled using the "collate" argument (29). Guided functional annotation of *de novo* scaffolds was performed by employing the Prokka v1.14.6. and Mtb H37Rv GenBank (Accession: GCF_000195955.2) was used as a reference with the "--proteins" argument (30). Gene Repertoires were determined using Panaroo v1.3.4. (31). Gene Ontology (GO) enrichment analysis was performed using the DAVID tool (https://david.ncifcrf.gov/).

## Pan-GWAS and statistical analysis

The Benjamin-Hochberg (BH) test was applied to determine the association of genes to drug-sensitive and drug-resistant Mtb clinical isolates using Scoary v1.6.16. (32). The genes qualifying the criteria of the BH adjusted $P$-value < 0.01 and $Log_{10}$ odds ratio (> 0.5 or <−0.5) were considered to have significantly perturbed associations either with drug-resistant or drug-sensitive isolates. Due to less number of drug-resistant Mtb isolates in the Netherland WGS data set, a BH-adjusted $P$-value of <0.05 was selected, and $log_{10}OR$ score threshold was not set to identify differentially present genes in the drug-resistant and drug-sensitive clinical Mtb isolates. The differentially present gene list was extracted from the reconstructed pangenome using "seqtk subseq" command and subjected to BLAST analysis against the Mtb H37Rv reference genome (Refseq accession Number: NC_000962.3). Genes that showed ≥95% identity score and subject to query length ratio ≥0.95–1.0 were considered as BLAST positive. BLAST results were used for annotating the gene clusters with locus tags for result interpretation. GraphPad Prism v8.0.2. and MS Excel (2016 home edition) were used for data analysis and representation.

## RESULTS

### Mtb WGS data analysis from TB-endemic region, filtering and population structure

The clinical Mtb isolates ($n$ = 2,601) used in this study were reported from India ($n$ = 2,232), China ($n$ = 201), Pakistan ($n$ = 80), and Zambia ($n$ = 88). Based on TBprofiler analysis, the total Mtb isolates were sub-grouped as drug sensitive ($n$ = 863), rifampicin resistant ($n$ = 90), isoniazid resistant ($n$ = 147), mono/poly-resistant ($n$ = 117), MDR ($n$ = 465), preXDR ($n$ = 883), and XDR ($n$ = 36). The classification of isolates into drug-resistant categories (MDR, XDR, and pre-XDR) was done according to WHO 2020 recommendations (https://www.who.int/publications/i/item/9789240018662). In 76 Mtb isolates, we detected more than one lineage of Mtb, indicating strain mixing, and the rest showed negligible signatures of strain mixing ($n$ = 2,525, k1) (Fig. 1A; Fig. S1A). Based on metagenome analysis, a subset of samples ($n$ = 365) had >5% of total alignment with non-Mtb organisms (that included Non-tuberculous *Mycobacteria* viz., *M. kansasii, M. chimera, M. marinum*, etc. and *Actinobacteria, Proteobacteria, Firmicutes, Chlamydiae, Crenarcheota* bacterial species) and the rest ($n$ = 2,236, k2) had ≥95% alignment specifically with MTBC species in the Minikraken database (Fig. 1A; Fig. S1B). Additional parameters like Mtb genome size (~4,411,532 bp), GC % (~65), and outliers having too many mismatches compared to the reference genome were used for filtering the scaffolded assemblies. Scaffolded *de novo* assemblies ($n$ = 2,466, k3) had GC% > 62, N50 >3,999,999, genome fraction relative to Mtb H37Rv > 95%, and mismatches per 100 kilobases < 100 (Fig. S1C). One sample failed to undergo scaffolding and was excluded. MD5Checksum of annotated genomes (GFF format) revealed that 2,600 annotated genomes were non-duplicates (k4). Finally, a set of 2,184 samples, qualified all data

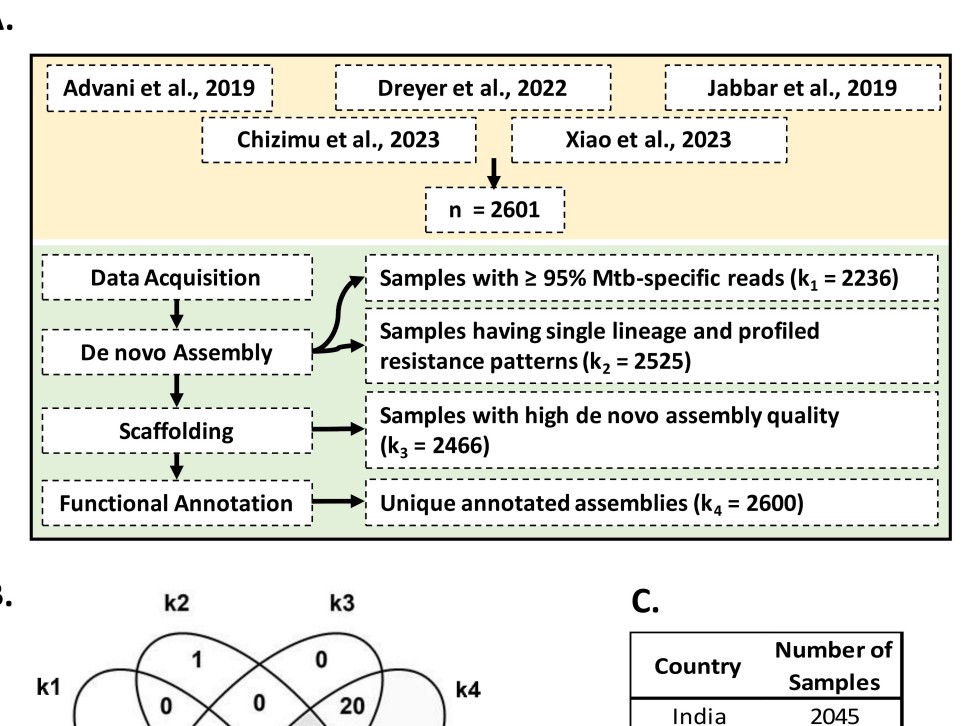

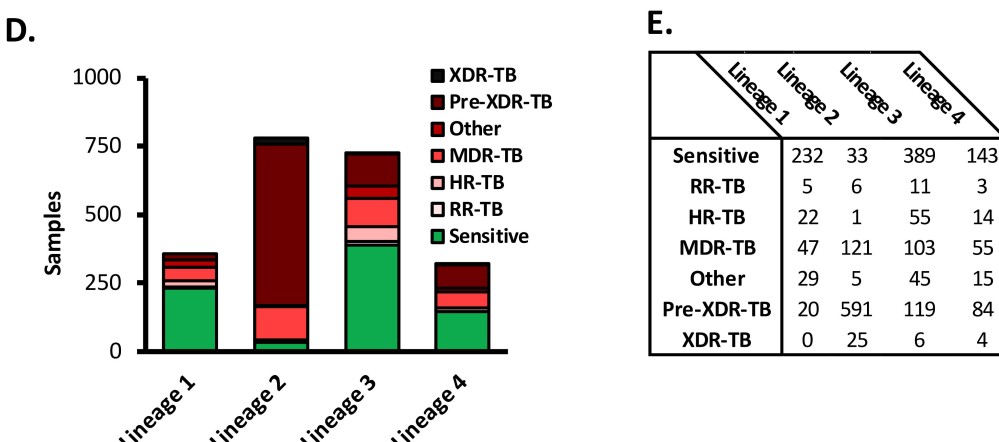

**FIG 1** Whole genome sequencing (WGS) data filtering and population structure of clinical *Mycobacterial tuberculosis* isolates included in the study. (A) Workflow employed to analyze publicly available WGS data. The data analysis consisted of data acquisition, assembling, DR profiling, lineage determination, metagenome detection, scaffolding, *de novo* assembly QC, functional annotation, and removal of duplicate GFF files. 2,601 samples were downloaded for data analysis. Sample in the sets $k_{1-4}$ successfully underwent pre-processing and showed high quality in one of the assessed metrics. (B) Venn diagram showing 2,184 samples were found common in $k_{1-4}$ and were subsequently subjected to gene repertoire analysis and Pan-GWAS. (C) The high-quality sample set ($n$ = 2184) consisted of Mtb isolates from four TB-endemic countries (India, Pakistan, Zambia, and China) with varying drug resistance profiles and lineages (highlighted in D and E). DR: drug resistant; n: the total number of samples downloaded from NCBI-SRA; $k_1$: number of samples passing >95% of reads aligning with MTBC species; $k_2$: number of samples having successfully characterized with no evidence of mixed lineages; $k_3$: number of samples that passed *de novo* quality metrics; $k_4$: number of samples passing deduplication based on MD5checksum of GFF files; XDR: extensively drug resistant; MDR: multidrug resistant; HR: isoniazid resistant; RR: rifampicin resistant.

filtering steps (DR profiling and strain mixing determination, Metagenome detection, *De novo* QC, and Md5Checksum-based de-duplication of annotated genomes) (Fig. 1B). We considered this data set as high quality and used it subsequently for Pangenome reconstruction and Pan-GWAS (Table S1).

The high-quality sample set ($n = 2184$) consisted of samples from India ($n = 2045$), China ($n = 9$), Pakistan ($n = 62$), and Zambia ($n = 68$) (Fig. 1C). A majority (95.8%) of Lineage 2 Mtb isolates and only 55%, 46.6%, and 34.6% of Lineage 4, Lineage 3, and Lineage 1 were drug resistant, respectively, in the high-quality sample set. Approximately 64% ($n = 1,386$) of Mtb isolates were drug resistant (25 Rif resistant, 95 isoniazid-mono resistant, 326 MDR, 94 mono/poly drug resistant, 814 preXDR, and 35 XDR samples), and 36% ($n = 798$) were classified as drug sensitive in the high-quality sample set (Fig. 1D and E).

## Pan-GWAS analysis reveals gene repertoire differences in drug-resistant and drug-sensitive Mtb clinical isolates

In total, a set of 4,893 genes were found in these 2,184 high-quality annotated Mtb genomes. Phylogenetic analysis grouped these samples into four separate clades corresponding to the four lineages (Fig. 2A). GR analysis of the qualified sample set using Panaroo showed 3,784 core genes (present in >99% of isolates), 123 softcore genes (present in 95%–98% of isolates), 224 shell genes (present in 15%–94% of isolates), and 762 cloud genes (present in >0%–14% isolates) (Fig. 2B). The majority of the identified pangenome (4,893 genes) belonged to the core (77.3%) category, and the rest could be classified as softcore (2.5%), shell (4.6%), and cloud (15.6%) genes. A set of 187 genes was significantly associated with either drug resistance or drug sensitivity status (BH adjusted *P*-value < 0.01). Among these, 115 genes aligned to multiple regions of Mtb H37Rv because of their sequence redundancy and were excluded from further analysis. Out of the rest 72 genes that showed a single hit after pairwise alignment, 39 genes showed a negative association with drug resistance (absent in drug-resistant Mtb clinical isolates) while 33 genes showed a positive association (present in drug-resistant Mtb clinical isolates). 13 out of 72 genes having significant association with either drug resistance or sensitivity status were identified as genes that constitute known regions of differences (RD) of Mtb, as per RDScan database (https://github.com/dbespiatykh/RDscan/blob/master/resources/RD.bed). These genes included Rv0071-73, Rv1355c, Rv1672c-73c, Rv1967, Rv1979c, Rv2816c-19c, and Rv3467.

In the 72 genes, we observed 11 genetic islands, each consisting of more than one tandem gene with respect to Mtb H37Rv. Six of these genetic islands (Rv0071-73, Rv1573-85c; Rv1672c-73c, Rv1760c-62c, Rv2816c-19c, and Rv3855-56c) showed a negative association with drug resistance (absent in drug-resistant Mtb clinical isolates) and five genetic islands (Rv0393-94c, Rv1787-88, Rv2318A-19c, Rv2645-46, and Rv2652c-59c) showed a positive association with drug resistance status (i.e., absent in drug-sensitive Mtb clinical isolates) (Tables 1 and 2).

Many yet-to-be-characterized hypothetical gene clusters were observed to have significant associations with one of the groups (R or S) viz., Rv0393 (R: Resistance), Rv0394c (R), Rv0963c (S: Sensitive), Rv0968 (R), Rv1761c-62c (S), Rv1765c (S), Rv2016 (R), Rv3467 (R), and Rv3856c (S). Phage protein genes (Rv1573c-85c) were observed to be negatively associated with drug resistance (absent in drug-resistant Mtb clinical isolates) and genes encoding prophage proteins (Rv2655c-59c) showed a positive association with drug resistance status (present in drug-resistant Mtb clinical isolates). After sorting the genes in the ascending order of $Log_{10}OR$ scores, Rv3855, Rv2765, Rv3830c, Rv0071, Rv0072, and Rv0073 were found as top six genes with most negative association values, whereas Rv3919c, Rv1844c, Rv1355c, Rv3383c, and Rv1371 were found as bottom-most five genes having most positive values. Apart from the genes with extreme $Log_{10}OR$ association scores, we observed negative and positive associations with drug resistance status of CRISPR-genes (Rv2816c-19c) and toxin-antitoxin genes (Rv2231A: VapC16 toxin and Rv2653c-54c), respectively, with high significance. The genes with significant

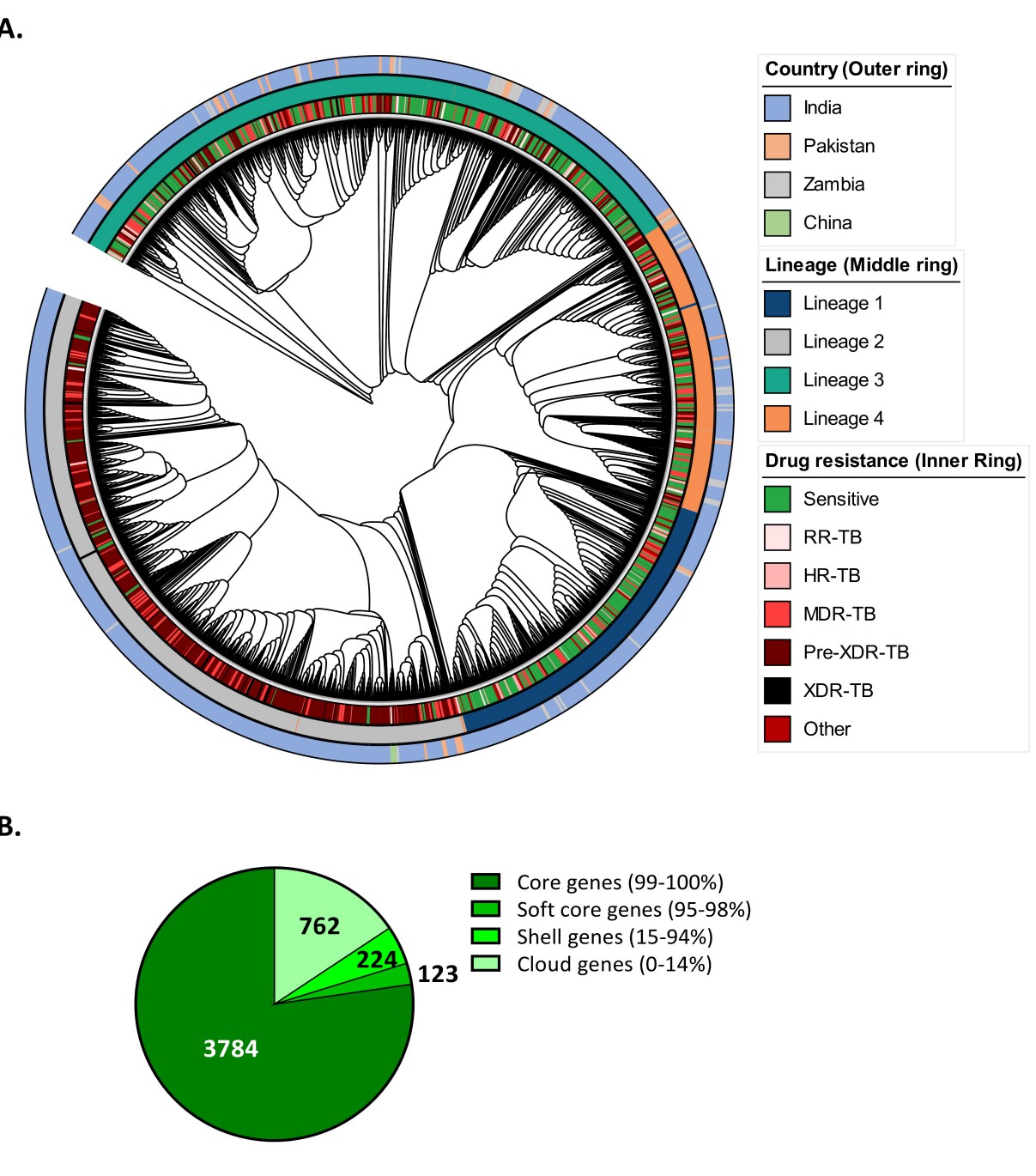

**FIG 2** Phylogenetic analysis and summary statistics of gene repertoire (GR) analysis. (A) Phylogenetic dendrogram of the high-quality sample set ($n = 2,184$) was used for the GR analysis. The accessory gene alignment in Newick format was visualized in iTOL online tool. The dendrogram shows four Mtb lineages as separate clades. (B) Summary statistics of gene repertoire analysis.

association either with drug resistance or drug sensitivity status are shown in Tables 1 and 2.

**TABLE 1** Genes negatively associated with drug resistance status of Mtb clinical isolates (BH-adjusted *P*-value: <0.01 and $Log_{10}$ odds ratio <−0.5)[a]

| Locus tag | Product | $Log_{10}$ OR |
|---|---|---|
| Rv3855 | HTH-type transcriptional repressor EthR | −2.13 |
| Rv3856c | hypothetical protein | −2.08 |
| Rv2765 | hydrolase | −1.64 |
| Rv3830c | TetR family transcriptional regulator | −1.42 |
| **Rv0071** | **Maturase** | **−1.41** |
| **Rv0073** | **Glutamine ABC transporter ATP-binding protein** | **−1.40** |
| **Rv0072** | **Glutamine ABC transporter permease** | **−1.37** |
| **Rv2816c** | **CRISPR-associated endoribonuclease Cas2** | **−1.30** |
| **Rv2818c** | **CRISPR-associated protein Csm6** | **−1.29** |
| **Rv2819c** | **CRISPR type III-associated RAMP protein Csm5** | **−1.24** |
| **Rv2817c** | **CRISPR-associated endonuclease Cas1** | **−1.04** |
| **Rv1673c** | **Hypothetical protein** | **−0.85** |
| **Rv1672c** | **Integral membrane transport protein** | **−0.84** |
| **Rv1967** | **Mce family protein Mce3B** | **−0.80** |
| Rv1038c | ESAT-6 like protein EsxJ | −0.80 |
| Rv1583c | Phage protein | −0.78 |
| Rv1582c | Phage protein | −0.77 |
| Rv1581c | Phage protein | −0.77 |
| Rv1578c | Phage protein | −0.77 |
| Rv1573 | Phage protein | −0.77 |
| Rv1576c | Phage capsid protein | −0.76 |
| Rv1580c | Phage protein | −0.76 |
| Rv1579c | Phage protein | −0.76 |
| Rv1575 | Phage protein | −0.75 |
| Rv1585c | Phage protein | −0.75 |
| Rv1527c | Polyketide synthase | −0.69 |
| Rv1760 | Diacylglycerol acyltransferase | −0.67 |
| Rv1762c | Hypothetical protein | −0.65 |
| Rv1758 | Cutinase | −0.64 |
| Rv1213 | Glucose-1-phosphate adenylyltransferase | −0.64 |
| Rv1761c | Hypothetical protein | −0.63 |
| Rv1765c | Hypothetical protein | −0.60 |
| Rv1557 | Transmembrane transport protein MmpL6 | −0.60 |
| Rv2319c | Universal stress protein | −0.60 |
| Rv3737 | Transmembrane protein | −0.58 |
| Rv1266c | Serine/threonine-protein kinase PknH | −0.57 |
| Rv3515c | Acyl-CoA synthetase | −0.56 |
| Rv3798 | Insertion sequence element IS1557 transposase | −0.54 |
| Rv0963c | Hypothetical protein | −0.51 |

[a]The table shows genes (locus tag), their products, and the value of $Log_{10}$OR, indicating the direction of the association. The loci marked in bold text are from RD as per the RDScan database. Abbreviations: RD: regions of difference; BH: Benjamin Hochberg; OR: odds ratio; S: sensitive; R: resistant.

## Gene ontology analysis reveals a compromised bacterial immune system in drug-resistant clinical Mtb isolates

DAVID GO analysis showed enrichment of biological processes viz., antiviral defense (GO0051607: Rv2818c, Rv2817c, Rv2819c, Rv2816c, and KW0051: Rv2818c, Rv2817c, Rv2819c, and Rv2816c), endonuclease activity (KW0255: Rv2818c, Rv2817c, and Rv2816c), nuclease activity (KW0540: Rv2818c, Rv2817c, and Rv2816c), and hydrolases (KW0378: Rv3856c, Rv2818c, Rv2765, Rv2817c, Rv1582c, Rv1758, and Rv2816c) in the drug-sensitive clinical Mtb isolates (Fig. 3A). It indicates a compromised bacterial immune system against invading phages in drug-resistant Mtb isolates. GO analysis

TABLE 2 Genes positively associated with drug resistance status of Mtb clinical isolates (BH-adjusted *P*-value < 0.01 and $Log_{10}$ odds ratio > +0.5)[a]

| Locus tag | Product | $Log_{10}OR$ |
|---|---|---|
| Rv3919c | 16S rRNA (guanine(527)-N (7))-methyltransferase RsmG | 2.22 |
| Rv1844c | 6-phosphogluconate dehydrogenase | 1.79 |
| **Rv1355c** | **Molybdopterin biosynthesis protein MoeY** | **1.40** |
| Rv2016 | Hypothetical protein | 1.39 |
| Rv3383c | Polyprenyl synthetase IdsB | 1.32 |
| Rv1371 | Membrane protein | 1.31 |
| Rv1730c | Penicillin-binding protein | 1.28 |
| Rv2318 | Sugar ABC transporter substrate-binding lipoprotein UspC | 1.27 |
| Rv0840c | Proline iminopeptidase | 1.13 |
| Rv0968 | Hypothetical protein | 1.05 |
| Rv1500 | Glycosyltransferase | 1.02 |
| Rv2231A | Ribonuclease VapC16 | 1.00 |
| Rv0648 | Alpha-mannosidase | 0.98 |
| Rv1255c | HTH-type transcriptional regulator | 0.84 |
| Rv0393 | Hypothetical protein | 0.69 |
| Rv1787 | PPE family protein PPE25 | 0.65 |
| Rv0394c | Hypothetical protein | 0.62 |
| Rv2653c | Toxin | 0.62 |
| Rv2645 | Hypothetical protein | 0.62 |
| Rv2656c | Prophage protein | 0.62 |
| Rv2658c | Prophage protein | 0.62 |
| Rv2657c | Prophage protein | 0.62 |
| Rv2659c | Prophage integrase | 0.61 |
| Rv2646 | Probable integrase | 0.61 |
| Rv2655c | Prophage protein | 0.61 |
| Rv2650c | Prophage protein | 0.60 |
| Rv1577c | Phage prohead protease | 0.58 |
| Rv2652c | Prophage protein | 0.58 |
| Rv2654c | Antitoxin | 0.57 |
| Rv1788 | PE family protein PE18 | 0.56 |
| **Rv3467** | **Hypothetical protein** | **0.54** |
| Rv3020c | ESAT-6 like protein EsxS | 0.53 |
| **Rv1979c** | **Permease** | **0.52** |

[a]The table shows genes (locus tag), their products, and the value of $Log_{10}OR$, indicating the direction of the association. The loci marked in bold text are from RD as per the RDScan database. Abbreviations: RD: regions of difference; BH: Benjamin Hochberg; OR: odds ratio; S: sensitive; R: resistant.

also revealed the presence of DNA binding domains (viz., HTH and HNH) in 17 out of 72 genes with perturbed associations potentially taking part in DNA modification or gene regulation. Out of 17 genes with DNA-binding domains, 12 genes had Nuclease, Primase, and Helicase activities that can take an active part in DNA modification. Out of 12 genes encoding DNA-modifying enzymes, 10 genes forming three genetic islands were observed. Island 1 consisted of Rv2646-59c, Island 2 consisted of Rv2816c-19c, and Island 3 consisted of Rv3855-56c (Fig. 3B). Most of the genes encoding DNA-modifying enzymes (Rv0071, Rv1582c, Rv1765c, Rv2646, Rv2657c-59c, Rv2816c-19c, Rv3467, Rv3798, and Rv3830c) have insertion sequences and/or repetitive DNA in close proximities (< 4 kb).

## Pilot Pan-GWAS comparing gene repertoires of drug-resistant Mtb in TB endemic and non-endemic regions

The clinical Mtb isolate WGS data (*n* = 1,130) from the Netherlands consisted of 914 (80.7%) drug-sensitive and 219 (19.3%) drug-resistant clinical Mtb isolates (Fig. S2A).

**A.**

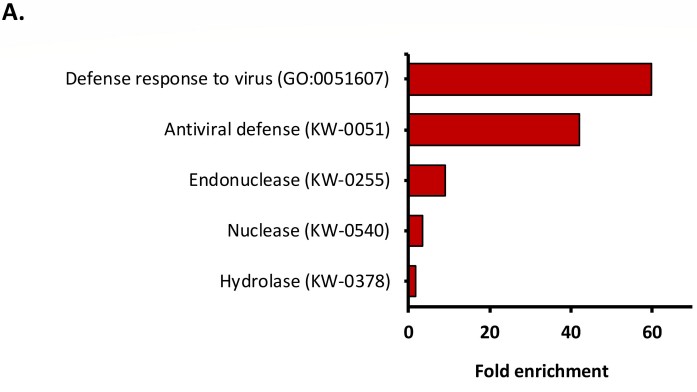

**B.**

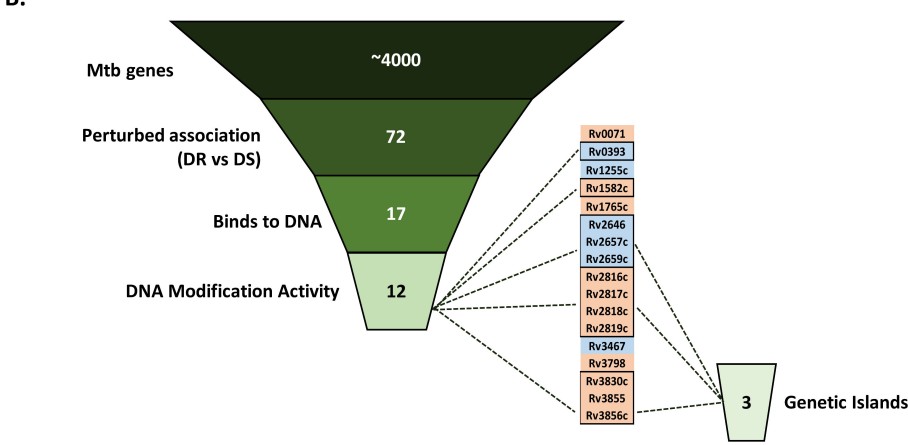

**C.**

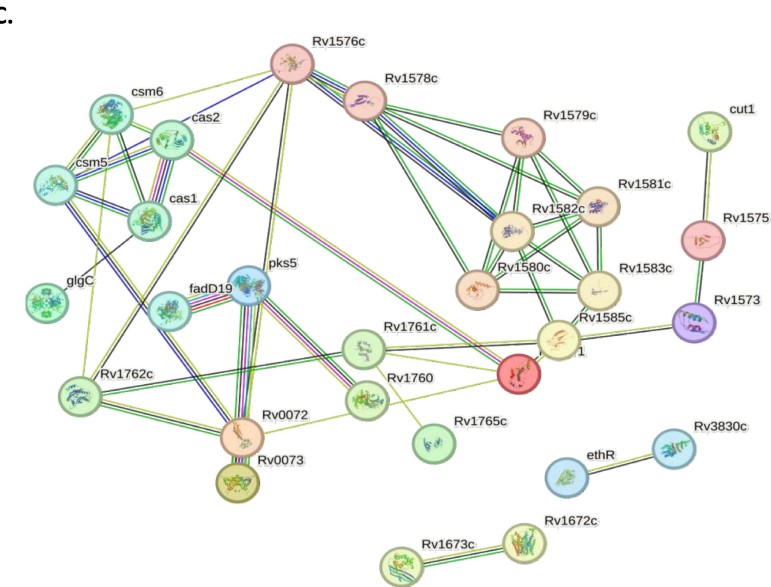

**FIG 3** Gene ontology clustering and relationships between deregulated genes. (A) Gene ontology clustering revealed the enrichment of genes involved in inter-related GO biological processes (defense response to invading viruses, symbionts, and other organisms). (B) Identifying genes potentially taking part in DNA modification. Pan-GWAS revealed 72 highly significant gene associations with drug-resistant and drug-sensitive isolates. 17 out of 72 genes encode domains that bind to DNA. 12 genes out of 17 genes with DNA-binding domains were found to have DNA modification activities. Three genetic islands were identified containing tandem genes. (C) StringDB network analysis revealed three clusters of genes having variable interactions: gene neighborhood (green), gene fusions (red), gene co-occurrence (blue), text mining (yellow), co-expressing (black), and protein homology (light blue).

From this, 1,126 (k1) samples had negligible contamination, 1,129 (k2) were of pure lineage, 1,130 (k3) could be assembled with high quality, and all 1,130 (k4) resulting annotated GFF files were unique (Fig. S2B through D). A set of 1,121 samples qualified all quality control steps and were used for pangenome reconstruction and Pan-GWAS analysis (Table S2). A set of 74 genes were differentially present in the drug-resistant and drug-sensitive Mtb clinical isolates reported from the Netherlands (Table S3).

Irrespective of the origin of the clinical Mtb isolates, either from endemic or non-endemic regions, 21 genes showed perturbed presence in the drug-resistant and drug-sensitive Mtb isolates (Fig. S3A and B). Among these, five genes (Rv0071, Rv0073, Rv1967, Rv2816c, and Rv2817c) were absent and Rv3919c was present in drug-resistant Mtb clinical isolates from both TB endemic and non-endemic regions analyzed. These genes may have a potential role in drug resistance development in Mtb. The rest 15 genes (Rv0394c, Rv1255c, Rv1967, Rv2645–46, Rv2650c, Rv2652c–59c, and Rv2818c) showed discordant patterns between TB endemic and non-endemic regions (Fig. S3C).

## DISCUSSION

Understanding the emergence of drug resistance in Mtb is quintessential to predicting its future evolutionary path. Under selective pressure, Mtb acquires mutations in specific genes that contribute to drug resistance development (3). These mutations can be used as markers for diagnosing the drug resistance status of Mtb clinical isolates by molecular tools. Many databases such as TBDreamDB, WHO mutation catalog, and PhyResSE provide the details of such drug resistance-associated mutations (33–35). Apart from these mutations, larger deletions have contribute to the development of drug resistance emergence in Mtb. These deletions may occur upon *Mycobacteriophage*-Mtb encounters and intragenomic recombination events (9, 10, 36). The Mtb genome houses a plethora of intermittently and continuously dispersed repetitive sequences such as insertion sequences and direct long and short repeats. These redundant genetic elements of Mtb are analyzed by various molecular typing tools such as IS6110 Restriction Fragment Length Polymorphism (IS6110-RFLP), Spoligotyping, Mycobacterial Interspersed Repetitive Unit-Variable Number of Tandem Repeats (MIRU-VNTR), and polymorphic GC-rich repetitive sequence Restriction Fragment Length Polymorphism (PGRS-RFLP) (36). These redundant genetic elements often present themselves in close proximities to each other and participate in genetic rearrangements or deletions in specific Mtb isolates (37–39). Mtb genome also houses genes such as IS6110 (insertion sequence and transposase), Rv3798 (probable transposase as per Mycobrowser), and RD1 region having genes with transposase-like activity (40). Factors like *Mycobacteriophages*-Mtb encounters, the presence of redundant genetic elements, and transposase genes make Mtb vulnerable to natural recombination events that can cause gene duplications and partial or complete deletion of genetic islands consisting of more than one non-essential gene. Many of these genetic islands serve as regions of differences (RD) and aid in the identification of infecting strains in surveillance and epidemiological studies and are also involved in conferring virulence in Mtb (41). Specific genes such as those that form membrane proteins and Secretome (e.g., RD155 genes including *eccCb1, PE35, esxB, esxA, eccD1, espK,* and P1cA-C) are reported to be associated with higher virulence in Mtb (41, 42). Based on these factors, we speculated that the genes with differential presence-absence patterns can potentially provide clues to understanding the evolution and emergence of drug resistance in Mtb.

In the present study, we conducted Pan-GWAS using publicly available WGS data of Mtb clinical isolates reported from TB-endemic countries (India, Pakistan, Zambia, and China). The presence of non-Mtb metagenomes in unprocessed WGS data can introduce foreign genes and affect the accuracy of the Pan-GWAS results so samples having signatures of non-Mtb metagenomes were excluded. WGS data set ($n$ = 2,184), which had high de-novo assembly quality and contained unique samples, was used for pangenome reconstruction and subsequent Pan-GWAS (Fig. 1). A majority (>95%) of the lineage 2 samples showed drug-resistant profiles (Fig. 1D; Fig. S1A) corroborating earlier

reports (43–45). GR analysis identified 3784 genes that form the core Mtb genome, which is within the reported range (15).

After computation of $Log_{10}OR$ values to determine the direction of association of specific genes having high significance (BH adjusted $P$-value < 0.01), we observed many intricacies that can be further investigated to understand the mechanistic aspects of drug resistance development as well as associated virulence. For example, the observed pattern in phage (negatively associated with drug resistance) and prophage proteins (positively associated with drug resistance) in this study in Mtb clinical isolates may be responsible for the previously known complex relationship between virulence and drug resistance observed in various bacteria, including in Mtb (46). A positive association of Toxin-Antitoxin genes (Rv2231A, Rv2653c, and Rv2654c) with drug resistance was observed. These Mtb genes are reported to be involved in the induction of dormancy, drug tolerance, and formation of persister Mtb population (47).

Through phylogenomic analysis, HGT has been identified as a driving force in the evolution of ancestral *Mycobacteria*. The Mtb genes with identical presence-absence patterns in certain genetic islands indicate that the constituting genes are perhaps acquired simultaneously in now-extinct ancestral *Mycobacterium prototuberculosis* through historical HGT or phage infection events (48). These genes were found to be co-expressing (based on the StringDB analysis). Upon analyzing the highly significant (BH adjusted $P$-value < 0.01) genes having extreme $Log_{10}OR$ scores, strong corroboration with existing literature was found. Rv3855, Rv2765, Rv3830c, Rv0071, Rv0072, and Rv0073 were found as the top six genes with the most negative association scores, meaning they are absent in a significant proportion of drug-resistant isolates. Rv3855 ($Log_{10}OR$ = −2.1, HTH-type transcriptional repressor EthR), a repressor of ethionamide-activating gene Rv3854, and its absence in drug-resistant isolates might not directly cause ethionamide drug resistance but could increase the tolerance to ethionamide. This tolerance could offer a survival advantage, potentially contributing to the evolution of drug resistance-like phenotype (49). Mutations in the upstream region of Rv2765 ($Log_{10}OR$ = −1.6, hydrolase) are previously known to be associated with ethambutol drug resistance (50). The absence of Rv3830c ($Log_{10}OR$ = −1.4, TetR family transcriptional regulator), which is a repressor that binds to the regulatory region of various multidrug efflux pumps, can potentially cause overexpression of its target multidrug efflux pump genes in the drug-resistant Mtb clinical isolates (51). Rv0071 ($Log_{10}OR$ = −1.4, maturase), Rv0072 ($Log_{10}OR$ = −1.3, glutamine ABC transporter permease), and Rv0073 ($Log_{10}OR$ = −1.4, glutamine ABC transporter ATP-binding protein) constitute RD105 as per RDScan database (52). The deletion of RD105 is already known to be associated with drug resistance development to multiple anti-TB drugs corroborating our findings (53).

Similarly, Rv3919c, Rv1844c, Rv1355c, Rv3383c, and Rv1371 were found to have the most positive $Log_{10}OR$ scores indicating a positive association with drug-resistant Mtb isolates, which means that these genes are present in a significant number of drug-resistant isolates. Rv3919c ($Log_{10}OR$ = +2.2, 16S rRNA (guanine(527)-N (7))-methyltransferase RsmG) is previously known to be associated with low-level resistance to streptomycin (54). Rv1844c ($Log_{10}OR$ = +1.79, 6-phosphogluconate dehydrogenase) is also previously known to be associated with Isoniazid resistance (55, 56). Rv1355c ($Log_{10}OR$ = +1.4, molybdopterin biosynthesis protein) showed a positive association with drug resistance. Rv1355c and the downstream gene Rv1356c are co-expressing genes (as per StringDB). Rv1356c is already known to have significantly lower protein expression in drug-sensitive Mtb isolates (57). Based on these factors and their tandem occurrence on the genome, Rv1355c and Rv1356c may have a relationship like that of a genetic island and an operon. Rv3383c ($Log_{10}OR$ = +1.3, polyprenyl synthetase IdsB) is known to have pyrazinamide resistance-associated mutation hotspots (58). A GWAS study previously revealed an association of Rv1371 ($Log_{10}OR$ = +1.3, a membrane protein) with resistance to drugs of multiple classes, including ethambutol, injectables, ethionamide, delamanid, and linezolid, corroborating with our findings (59).

Important to highlight is the seemingly critical role of CRISPR-associated genes ($Log_{10}OR = -1.3$ to $-1.04$) that showed a negative association with drug resistance. The absence of Rv2816c-17c in most Lineage 2 Mtb isolates is known and its absence can have a cumulative effect on the impaired DNA repair in drug-resistant Mtb isolates. The study also highlighted the potential role of the insertion sequence present upstream of Rv2816c-17c-18c-19c in genomic deletions (60). Also, overexpression of Rv2816c is previously known to decrease the drug susceptibility in *M. smegmatis,* whereas Rv2816c-knockout strains are also known to impart drug tolerance and drug resistance-like phenotype (61–63). Wang et al. recently discussed the association of the absent CRISPR system with drug resistance in *Klebsiella pneumoniae* (64). Similarly, its absence is known to be associated with drug resistance in *Shigella* also (65). The independent pilot Pan-GWAS analysis of the Mtb WGS data set from a TB non-endemic region, that is, the Netherlands also showed the absence of CRISPR genes (Rv2816c and Rv2817c), Rv0071, Rv0073, and the presence of Rv3919c in drug-resistant Mtb clinical isolates corroborating the observations from the TB-endemic countries. The observed discordance results of Pan-GWAS conducted in TB-endemic and non-endemic regions could be due to the lower number of drug-resistant Mtb isolates from the non-endemic country and majorly from the differential selection pressures in geographically isolated places, which could alter evolutionary trajectories due to allopatric evolution. Many genes, including the hypothetical ones, showed strong association with drug resistance development in Mtb. These genes may not have a direct causation of drug resistance, therefore, elucidating their roles with respect to the emergence of drug resistance can potentially enrich our understanding of drug resistance evolution in Mtb.

## Conclusions

The present study highlights differences in the genetic makeup of drug-resistant and drug-sensitive Mtb clinical isolates at the pangenome scale. In addition, the deletion of various characterized and hypothetical genes, such as CRISPR genes (Rv2816-19c), was identified in the clinical drug-resistant Mtb clinical isolates which might explain the evolutionary trajectory followed by these difficult-to-treat pathogens. Therefore, experimentally dissecting the role of these gene sets selectively present or absent in drug-resistant Mtb clinical isolates at molecular and protein levels may provide insights into the emergence of drug resistance development.

## ACKNOWLEDGMENTS

R.K.N. acknowledges the Department of Biotechnology (DBT), India for Funding (Grant ID: National Network Project of National Institute of Immunology, New Delhi -[40267]). Nidhi Yadav and Ashish Gupta from the Translational Health Group, ICGEB, New Delhi component are acknowledged for providing critical feedback and reviewing the manuscript.

## AUTHOR AFFILIATION

[1]Translational Health Group, International Center of Genetic Engineering and Biotechnology, New Delhi, India

## AUTHOR ORCIDs

Nikhil Bhalla  http://orcid.org/0000-0003-0584-5726
Ranjan Kumar Nanda  http://orcid.org/0000-0002-2163-4411

## FUNDING

| Funder | Grant(s) | Author(s) |
|---|---|---|
| Department of Biotechnology, Ministry of Science and Technology, India (DBT) | 40267 | Ranjan Kumar Nanda |

## AUTHOR CONTRIBUTIONS

Nikhil Bhalla, Conceptualization, Data curation, Formal analysis, Investigation, Methodology, Visualization, Writing – original draft, Writing – review and editing | Ranjan Kumar Nanda, Formal analysis, Funding acquisition, Investigation, Methodology, Project administration, Resources, Supervision, Writing – review and editing

## ADDITIONAL FILES

The following material is available online.

### Supplemental Material

**Supplemental material (Spectrum00527-24-S0001.pdf).** Fig. S1 to S3; Tables S1 to S3.

### Open Peer Review

**PEER REVIEW HISTORY (review-history.pdf).** An accounting of the reviewer comments and feedback.

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
