## [Reviewer comments · Microbiology Spectrum]

Microbiology Spectrum

Pangenome-wide association study reveals selective absence of CRISPR genes (Rv2816c-19c) in drug-resistant *Mycobacterium tuberculosis*.

Nikhil Bhalla and Ranjan Nanda

Corresponding Author(s): Ranjan Nanda, International Centre for Genetic Engineering and Biotechnology New Delhi

Review Timeline:

Submission Date:	February 27, 2024
Editorial Decision:	April 22, 2024
Revision Received:	May 7, 2024
Accepted:	May 31, 2024

Editor: Felix Toka

Reviewer(s): Disclosure of reviewer identity is with reference to reviewer comments included in decision letter(s). The following individuals involved in review of your submission have agreed to reveal their identity: Sudha Ramaiah (Reviewer #1); Nena A (Reviewer #2); Kaan Çeylan (Reviewer #4)

Transaction Report:

DOI: <https://doi.org/10.1128/spectrum.00527-24>

Re: Spectrum00527-24 (Pangenome-wide association study reveals selective absence of CRISPR genes (Rv2816c-19c) in drug-resistant Mycobacterium tuberculosis.)

Dear Dr. Ranjan Kumar Nanda:

Thank you for the privilege of reviewing your work. Below you will find my comments, instructions from the Spectrum editorial office, and the reviewer comments.

The manuscript needs revisions in the abstract, materials and methods, results, and discussion sections, as recommended by Reviewer #1. The authors should also consider depositing the raw sequences in a repository like the NCBI Sequence Read Archive (SRA).

Revision Guidelines

Sincerely,
Felix Toka
Editor
Microbiology Spectrum

Reviewer #1 (Comments for the Author):

Manuscript Title: Pangenome-wide association study reveals selective absence of CRISPR genes 1 (Rv2816c-19c) in drug-resistant Mycobacterium tuberculosis.

Recommendation: Major Revision

The study entitled "Pangenome-wide association study reveals selective absence of CRISPR genes 1 (Rv2816c-19c) in drug-resistant Mycobacterium tuberculosis." by Nikil Bhalla and Ranjan Kumar Nanda have studied and explored drug resistance development in Mycobacterium tuberculosis (Mtb) by analyzing Whole Genome Sequencing (WGS) data of clinical Mtb isolates. They identified genes CRISPR genes 1 (Rv2816c-19c) associated with drug-resistant Mtb isolates and found that drug-resistant isolates showed compromised bacterial immune systems and impaired DNA repair. The study suggests potential drug targets that could help in understanding host-pathogen interactions and can be potentially helpful in development of diagnostic tools. Few major queries need to be addressed before the study can be recommended for publication. Authors might consider the following suggestions.

1. The rationale of the study should be incorporated in the abstract focusing on why is this study important.
2. Add more keywords; remember that the selected keywords should accurately reflect the content of your paper and be relevant to the research community interested in your topic.
3. The samples size considered for the study have great difference. The authors are suggested to considered the same sample size from all the countries to make the study less biased.
4. In the results section (Page 6, lines 184 - 187), the authors talk about the pangenome with core, shell and cloud genes. Can the authors give their percentages for each category for comparison?
5. The authors should mention about the Function annotation and Pathway analysis. It is suggested to the authors that gene ontology terms associated with the interactome should be explained in detail. Provide all the relevant data for the enrichment performed.
6. In the discussion section Page No. 9; Line No. 284-287 provide the citation for the content. Similarly provide the citation from the existing literature for the content in Line 288-291.
7. The authors are suggested to include conclusion section separately in the manuscript focusing on the outcome of the study with future insights.
8. The author should consider redo some of the figures. Many figures show low picture quality or distorted. These figures may need better design and arrangement for comprehensive data presentation.
9. The authors are suggested use some referencing tool in order to include citation. It should be consistent throughout the manuscript.
10. Grammatical, language editing and spelling is highly recommended using professional software and/ or language experts.

Reviewer #2 (Comments for the Author):

The data can be collected from endemic and non-endemic locations

**Pangenome-wide association study reveals selective absence of CRISPR genes**
**(Rv2816c-19c) in drug-resistant *Mycobacterium tuberculosis*.**

**Nikhil Bhalla^{1,*}, Ranjan Kumar Nanda^{1,*}**

¹Translational Health Group, International Center of Genetic Engineering and Biotechnology,
New Delhi Component, Aruna Asif Ali Road, New Delhi, India.

***Corresponding author(s):**

Nikhil Bhalla (Ph.D.)

Email: nikhilbhalla94@gmail.com

**or,**

Ranjan Kumar Nanda (Ph.D.)

Email: ranjan@icgeb.res.in

Translational Health Group, International Centre for Genetic Engineering and Biotechnology,
New Delhi-110067, India

Telephone: +91-11-26741358

Fax: +91-11-26742316

**Abstract**

**Background:** Drug resistance development in *Mycobacterium tuberculosis* (Mtb) is
attributed to the acquisition of mutations in specific Mtb genes. The role of the differential
presence of non-essential genes in drug resistance development is relatively unexplored.

**Methodology:** Publicly available Whole Genome Sequencing (WGS) data of clinical Mtb
isolates (n=2601) from TB endemic countries was used. The Mtb WGS data was de novo
assembled, filtered for contamination, scaffolded into longer contigs, and functionally
annotated. For the Pangenome-Wide Association Study (Pan-GWAS), Benjamin Hochberg
test was applied to identify genes that are significantly associated with drug-resistant Mtb
isolates.

**Results:** From 2601 Mtb WGS data sets, 2184 (drug-resistant/sensitive: 1386/798) qualified
as high-quality. A set of 3784 core genes, 123 softcore genes, 224 shell genes, and 762 cloud
genes were identified. Sets of 33 and 39 genes showed a positive and a negative association
with drug-resistant isolates, respectively, with high significance (p-value < 0.01). Gene
ontology cluster analysis indicated a compromised bacterial immune system and impaired
DNA repair in drug-resistant compared to the sensitive Mtb isolates. Multidrug efflux pump
repressor genes (Rv3830c and Rv3855c) were also absent in the drug-resistant Mtb isolates.
The absence of CRISPR-associated genes is reported in other drug-resistant microbes, and a
similar pattern of Rv2816c-19c is observed in Mtb.

**Conclusions:** The study sheds light on Mtb genes involved in drug resistance emergence and
could be helpful in better understanding host-pathogen interactions, identification of novel
drug targets and development of diagnostic tools.

**Keywords:** Drug resistance, GWAS, *M. tuberculosis*, Gene repertoires.

**Importance**

The results from the present Pan-GWAS study comparing gene sets in drug-resistant and
drug-sensitive Mtb isolates revealed intricate presence-absence patterns of genes encoding
DNA binding proteins having gene regulatory as well as DNA modification and DNA repair
roles. Apart from the genes with known functions, we have identified some Uncharacterized
and Hypothetical genes that seem to have a significant role in drug resistance development in
Mtb. We have been able to extrapolate many findings of the present study with the existing
literature on drug-resistant Mtb isolates, further strengthening the relevance of the results
presented in this study. This study contributes to the foundation of future research on the drug
resistance development in Mtb.

**Introduction:**

According to the Global Tuberculosis Report 2023 by the World Health Organization
(WHO), approximately 410,000 new drug-resistant Tuberculosis (TB) cases were reported in
2022. TB has a higher prevalence in tropical regions, such as South Asia and Africa, that
mainly comprise poverty-stricken developing nations, compared to other parts of the world.
The causative agent of TB, i.e.,  *Mycobacterium tuberculosis* (Mtb), is a slow-growing
pathogen that mostly causes lower respiratory tract infections. Mtb has several intrinsic drug
resistance-conferring factors including thick cell wall, lipid-rich cell membrane, drug-
inactivating enzymes and drug target modification systems. Host-dependent selective
pressures such as incompliance, inappropriate dose of antibiotics, and factors like delayed
diagnosis lead to the accumulation of mutations in specific Mtb genes that result in
ineffective drug action (1,2,3,4). Despite the introduction of several novel and repurposed
drugs for the treatment of drug-resistant TB, unresponsiveness to new therapeutics is
increasingly being reported (5). Mtb's slow growth rate, ability to remain dormant for
decades, development of drug tolerance, and drug resistance also contribute to the relapse of
TB.

Drug resistance in Mtb is primarily caused by the acquisition of Single Nucleotide
Polymorphisms (SNPs) in the drug target genes, prodrug-activating genes, their promoter
regions, and other genes that are involved in the mechanism of action of respective drugs.
Unlike in other bacteria, Horizontal Gene Transfer (HGT) of the drug resistance-conferring
plasmids does not contribute to the development of drug resistance in Mtb (6–8). However,
Mtb is susceptible to infection by *Mycobacteriophages* and can also undergo natural intra-
genome recombination events (9,10). In a recent study, the insertion sequence IS6110 that
encodes a transposase is found to actively take part in transposition, thereby leading to
genetic deletions in an observable time frame of one year. IS6110 is also reportedly more
abundant in Lineage 2 (Beijing) Mtb isolates, which are widely known to be highly drug-
resistant (11,12). Events like gene deletions can directly contribute to the emergence of drug
resistance or indirectly by improving the fitness.

Genome-Wide Association Studies (GWAS) have been extensively used to identify drug
resistance-associated SNPs in Mtb, but its application at the pangenome level is limited (13).
Existing literature on Pan-GWAS of Mtb seems to be primarily focused on identifying the
genetic determinants taking part in higher prevalence, the site of infection

(Extrapulmonary/Pulmonary) and those forming inter-species diversity (14–16). A Pan-
GWAS study from 2018 identified 24 novel genetic signatures associated with drug resistance
using a sample set of 1595 from varying geographical regions (17). Another Pan-GWAS
study, also from 2018, was more focused on understanding the causation of atypical drug
resistance in Mtb isolates. In this, several unique genes, as well as intergenic regions, are
found to be exclusively associated with atypical drug resistance in Mtb compared to 145
other isolates with typical drug-resistant markers (18).

Given the limited literature on the pangenome of drug-resistant Mtb isolates and our
understanding of genomic structural variations that may arise upon drug resistance
development, we aimed to identify unique gene repertoires by analysing a publicly available
Mtb WGS data set from TB endemic countries. The identified gene sets in the present study
might be useful in developing region-specific diagnostic tools, identifying novel drug targets
and understanding the host-pathogen interactions of drug-resistant Mtb isolates.

**Methodology:**

**Data acquisition, De novo assembly, and Metagenome detection:** From existing reports
with corresponding NCBI-BioProjects: PRJNA879962, PRJEB41116, PRJEB32684,
PRJNA379070, and PRJEB29435, Mtb WGS data (n=2601) from India, China, Pakistan, and
Zambia was downloaded from the NCBI-Sequence Repository Archive (SRA) using fasterq-
dump v2.11.3 of the SRA toolkit (NCBI) (19–23). Megahit assembler v1.2.9. was used for
the de novo assembly of Mtb WGS data (24). Ragtag v2.1.0. was used for the scaffolding of
Megahit assemblies (25). Kraken2 v2.1.3. was used for the detection of metagenomes using
Minikraken database
(https://ccb.jhu.edu/software/kraken/dl/minikraken_20171019_8GB.tgz) (26). The quality of
de novo assemblies was determined using QUAST v5.2.0 (27).

**Drug resistance profiling, Gene Repertoire analysis, and Statistics:** For drug resistance
profiling and lineage identification of the Mtb isolates, the Megahit assemblies were used as
input for TBprofiler v5.0.0. (Database: n0599ccdEJody) and the output data was compiled
using the "collate" argument (28,29). Guided functional annotation of de novo scaffolds was
performed by employing the Prokka v1.14.6. and Mtb H37Rv GenBank (Accession:
GCF_000195955.2) was used as reference with "--proteins" argument (30). Gene Repertoires
were determined using Panaroo v1.3.4. (31). Gene Ontology clustering was performed using
DAVID (<https://david.ncifcrf.gov/>).

**Statistical analysis:** Benjamin-Hochberg (BH) test was applied for determining the
association of genes to drug-sensitive and drug resistant Mtb isolates using Scoary v1.6.16.
(32). The genes qualifying the criteria of the BH adjusted p-value (< 0.01) and Log_{10} Odds
ratio (> 0.5 or < -0.5) were considered to have significantly perturbed association either with
drug-resistance or drug-sensitive isolates. GraphPad Prism v8.0.2. and MS Excel (2016 home
edition) were used for data analysis and representation.

**Results:**

**WGS data analysis, filtering and population structure:** The clinical Mtb isolates (n=2601)
used in the present study were reported from India (n=2232), China (n=201), Pakistan
(n=80), and Zambia (n=88) (S2-Table 1). Based on TBprofiler analysis, the total Mtb isolates
were sub-grouped as drug-sensitive (n=863), rifampicin-resistant (n=90), isoniazid-resistant
(n=147), mono/poly-resistant (n=117), MDR (n=465), preXDR (n=883), and XDR (n=36).
The classification of isolates into drug-resistant categories (MDR, XDR, and pre-XDR) was
done according to WHO 2020 recommendations
(<https://www.who.int/publications/i/item/9789240018662>). In 76 Mtb isolates, we detected
with more than one lineage of Mtb, indicating strain mixing, and the rest showed negligible
signatures of strain mixing (n=2525, k1) (Figure 1-A and 1S-A). Based on Metagenome
analysis, a subset of samples (n=365) had $> 5\%$ of total alignment with non-Mtb organisms
(that included Non-tuberculous *Mycobacteria* viz., *M. kansasii*, *M. chimera*, *M. marinum*, etc
and *Actinobacteria*, *Proteobacteria*, *Firmicutes*, *Chlamydiae*, *Crenarcheota* bacterial species)
and the rest (n=2236, k2) had $\geq 95\%$ alignment specifically with MTBC species in the
Minikraken database (Figure 1A and S-1B).

Additional parameters like Mtb genome size (~4411532 bp), GC % (~65), and outliers having
too many mismatches compared to the reference genome were used for filtering the
scaffolded assemblies. Scaffolded de novo assemblies (n=2466, k3) had $\text{GC}\% > 62$, $\text{N50} >$
3999999 , genome fraction relative to Mtb H37Rv $> 95\%$, and mismatches per 100 kilobases
< 100 (Figure 1S-C). One sample failed to undergo scaffolding and was excluded.

MD5Checksum of annotated genomes (GFF format) revealed that 2600 annotated genomes
were non-duplicates (k4) (Figure 1A).

Finally, a set of 2184 samples, qualified all data filtering steps (DR profiling and strain
mixing determination, Metagenome detection, De novo QC, and Md5Checksum-based de-

duplication of annotated genomes) (Figure 1B). We considered this data set as high-quality
and subsequently used for Pan-GWAS.

The high-quality sample set (n=2184) consisted samples from India (n=2045), China (n=9),
Pakistan (n=62) and Zambia (n=68) (Figure 1C). A majority (95.8%) of Lineage 2 Mtb
isolates and only 55%, 46.6% and 34.6% of Lineage 4, Lineage 3 and Lineage 1 were drug
resistant, respectively, in the high-quality sample set. Approximately 64% (n=1386) of Mtb
isolates were drug-resistant (25 Rif-resistant, 95 isoniazid-mono-resistant, 326 MDR, 94
mono/poly drug-resistant, 814 preXDR and 35 XDR samples), and 36% (n=798) were
classified as drug-sensitive in the high-quality sample set (Figure 1D, 1E and S2-Table 2).
Phylogenetic analysis grouped these samples into 4 separate clades corresponding to the four
lineage (Figure 2A).

**Pan-GWAS analysis reveals gene repertoire differences in drug-resistant and sensitive**
**Mtb isolates:** In total, a set of 4893 genes were found in these 2184 high-quality annotated
Mtb genomes (Figure 2B). Gene repertoire analysis of the qualified sample set using Panaroo
showed 3784 core genes (present in >99 % of isolates), 123 softcore genes (present in 95-
98% of isolates), 224 shell genes (present in 15-94% of isolates), and 762 genes (present in
>0-14% isolates).Click or tap here to enter text.

A set of 187 genes was found to be significantly associated with either drug resistance or drug
sensitivity status (Benjamin Hochberg adjusted p-value < 0.01). Amongst these, 115 genes
aligned to multiple regions of Mtb H37Rv because of their sequence redundancy and were
excluded from further analysis. Out of the rest 72 genes that showed a single hit after
pairwise alignment, 39 genes showed a negative association with drug resistance (absent in
drug-resistant isolates), while 33 genes showed a positive association (present in drug
resistant isolates).

13 out of 72 genes having significant association with either drug resistance or sensitivity
status were identified as genes that constitute known region of differences (RD) of Mtb, as
197 per [RDScan](https://github.com/dbespiatykh/RDscan/blob/master/resources/RD.bed) database
(<https://github.com/dbespiatykh/RDscan/blob/master/resources/RD.bed>). These genes
included Rv0071-73, Rv1355c, Rv1672c-73c, Rv1967, Rv1979c, Rv2816c-19c and Rv3467.

In the 72 gene, we observed eleven genetic islands, each consisting of more than one tandem
gene with respect to Mtb H37Rv. Six of these genetic islands (Rv0071-73, Rv1573-85c;
Rv1672c-73c, Rv1760c-62c, Rv2816c-19c, and Rv3855-56c) showed negative association
with drug resistance (absent in drug-resistant isolates) and 5 genetic islands (Rv0393-94c,

Rv1787-88, Rv2318A-19c, Rv2645-46, Rv2652c-59c) showed positive association with drug
resistance status (i.e. absent in drug-sensitive isolates) (Table 1).

Many yet-to-be-characterized hypothetical gene clusters were observed to have significant
associations with one of the groups viz., Rv0393 (Resistance), Rv0394c (R), Rv0963c
(Sensitive), Rv0968 (R), Rv1761c-62c (S), Rv1765c (S), Rv2016 (R), Rv3467 (R), and
Rv3856c (S). Phage protein genes (Rv1573c-85c) were observed to be negatively associated
with drug resistance (absent in drug-resistant Mtb isolates) and genes encoding prophage
proteins (Rv2655c-59c) showed positive association with drug resistance status (present in
drug-resistant isolates). After sorting the genes in the ascending order of Log_{10}OR scores,
Rv3855, Rv2765, Rv3830c, Rv0071, Rv0072 and Rv0073 were found as top 6 genes with
most negative association values whereas Rv3919c, Rv1844c, Rv1355c, Rv3383c, Rv1371
were found as bottom-most 5 genes having most positive values.

Apart from the genes with extreme Log_{10}OR association scores, we observed negative and
positive association with drug resistance status of CRISPR-genes (Rv2816c-19c) and Toxin-
Antitoxin genes (Rv2231A: VapC16 toxin and Rv2653c-54c), respectively, with high
significance. The genes with significant association either with drug resistance or drug
sensitivity status are shown in Table 1A and 1B.

**Gene Ontology analysis:** DAVID gene ontology (GO) clustering of the genes showed
enrichment of biological processes viz., antiviral defense (GO 0051607 and KW0051),
endonuclease activity (KW0255), nuclease activity (KW0540), and Hydrolases (KW0378)
drug-sensitive Mtb isolates (Figure 3A). This indicates that the bacterial immune system is
compromised in most drug-resistant Mtb isolates.

[revised manuscript text omitted]

The Mtb genes with identical presence-absence patterns in certain genetic islands indicates
that the constituting genes are perhaps acquired simultaneously in now-extinct ancestral
*Mycobacterium prototuberculosis* through historical HGT or phage infection events. These
genes were subsequently found to be co-expressing (StringDB).

Upon analyzing the highly significant ($p\text{-value} < 0.01$) genes having extreme Log_{10}OR
scores, strong corroboration with existing literature was found. Rv3855, Rv2765, Rv3830c,
Rv0071, Rv0072 and Rv0073 were found as top 6 genes with most negative association
scores which means that they are absent in significant proportion of drug resistant isolates.
Rv3855 ($\text{Log}_{10}\text{OR}=-2.1$, HTH-type transcriptional repressor EthR), a repressor of
ethionamide-activating gene Rv3854 and its absence in drug resistant isolates might not
directly cause ethionamide drug resistance but could increase the tolerance to ethionamide.
This tolerance could offer a survival advantage, potentially contributing to the evolution of
drug resistance like phenotype (48). Mutations in the upstream region of, Rv2765
($\text{Log}_{10}\text{OR}=-1.6$, Hydrolase) are previously known to be associated with ethambutol drug
resistance (49). Absence of Rv3830c ($\text{Log}_{10}\text{OR}=-1.4$, TetR family transcriptional regulator),
which is a repressor that binds to the regulatory region of various multidrug efflux pumps can
potentially cause overexpression of its target multidrug efflux pump genes in the drug
resistant Mtb isolates (50). Rv0071 ($\text{Log}_{10}\text{OR}=-1.4$, Maturase), Rv0072 ($\text{Log}_{10}\text{OR}=1.3$,
Glutamine ABC transporter permease) and Rv0073 ($\text{Log}_{10}\text{OR}=-1.4$, Glutamine ABC
transporter ATP binding protein) constitute RD105 as per RDScan database (51). The deletion
of RD105 is already known to be associated with drug resistance development to multiple
anti-TB drugs corroborating our findings (52).

Similarly, Rv3919c, Rv1844c, Rv1355c, Rv3383c, and Rv1371 were found to have the most
positive Log₁₀OR scores indicating a positive association with drug-resistant Mtb isolates,
which means that these genes are present in a significant number of drug-resistant isolates.
Rv3919c (Log₁₀OR=+2.2, 16S rRNA (guanine(527)-N(7))-methyltransferase RsmG), is
previously known to be associated with low-level resistance to streptomycin (53). Rv1844c
(Log₁₀OR=+1.79, 6-Phosphogluconate dehydrogenase), is also previously known to be
associated with Isoniazid resistance (54,55). Rv1355c (Log₁₀OR=+1.4, Molybdopterin
biosynthesis protein) showed positive association with drug resistance. Rv1355c and the
downstream gene Rv1356c are co-expressing genes (as per StringDB). Rv1356c is already
known to have significantly lower expression in drug-sensitive Mtb isolates (56). Based on
these factors as well as their tandem occurrence on the genome, Rv1355c and Rv1356c can
be speculated to have a relationship like that of a genetic island and an operon. Rv3383c
(Log₁₀OR=+1.3, Polyprenyl synthetase IdsB), is known to have pyrazinamide resistance-
associated mutation hotspots (57). A GWAS study previously revealed an association of
Rv1371 (Log₁₀OR=+1.3, a membrane protein) with resistance to drugs of multiple classes,
including Ethambutol, injectables, Ethionamide, Delamanid and Linezolid, corroborating
with our findings (58).

Important to highlight is the seemingly critical role of CRISPR-associated genes (Log₁₀OR=-
1.3 to -1.04) that showed negative association with drug resistance. Absence of Rv2816c-17c
is most Lineage 2 Mtb isolates is previously known and that its absence can have a
cumulative effect on the impaired DNA repair in drug-resistant Mtb isolates (59). Also,
overexpression of Rv2816c is previously known to decrease the drug susceptibility in *M.*
*smegmatis* whereas Rv2816c-knockout strains are also known to impart drug tolerance and
drug resistance-like phenotype (59–62). Wang et al. recently discussed the association of the
absent CRISPR system with drug resistance in *Klebsiella pneumoniae* (63). Similarly, its
absence is also known to be associated with drug resistance in *Shigella* (64). Earlier reports
showed that the upstream region of Rv2816c-17c-18c-19c comprises insertion sequences and
direct repeats, which are vulnerable to insertion sequence-driven genome deletions (65).

Many genes including the hypothetical ones showed strong association with drug resistance
development in Mtb. These genes may not have a direct causation of drug resistance,
therefore, elucidating their roles with respect to the emergence drug resistance can potentially
enrich our understanding of drug resistance evolution in Mtb.

**Contributions:** NB conceptualized the study and analysed data. RKN provided guidance for
the execution of analysis. NB and RKN prepared the manuscript.

**Acknowledgement:** NB and RKN acknowledge Department of Biotechnology (DBT), India
for Funding (Grant ID: National Network Project of National Institute of Immunology, New
Delhi -[40267]). Nidhi Yadav and Ashish Gupta from TH Group, ICGEB, New Delhi
component are acknowledged for providing critical feedback and reviewing the manuscript.

**References**

- 1. Seung, K. J., Keshavjee, S., & Rich, M. L. (2015). Multidrug-Resistant Tuberculosis
and Extensively Drug-Resistant Tuberculosis. *Cold Spring Harbor perspectives in*
*medicine*, 5(9), a017863. <https://doi.org/10.1101/cshperspect.a017863>
- 2. Seung, K. J., Gelmanova, I. E., Peremitin, G. G., Golubchikova, V. T., Pavlova, V. E.,
Sirotkina, O. B., Yanova, G. V., & Strelis, A. K. (2004). The effect of initial drug
resistance on treatment response and acquired drug resistance during standardized
short-course chemotherapy for tuberculosis. *Clinical infectious diseases*, 39(9), 1321–
1328. <https://doi.org/10.1086/425005>
- 3. Gygli, S. M., Borrell, S., Trauner, A., & Gagneux, S. (2017). Antimicrobial resistance
in *Mycobacterium tuberculosis*: mechanistic and evolutionary perspectives. *FEMS*
*microbiology reviews*, 41(3), 354–373. <https://doi.org/10.1093/femsre/fux011>
- 4. Poulton, N. C., & Rock, J. M. (2022). Unraveling the mechanisms of intrinsic drug
resistance in *Mycobacterium tuberculosis*. *Frontiers in cellular and infection*
*microbiology*, 12, 997283. <https://doi.org/10.3389/fcimb.2022.997283>
- 5. Pym, A. S., Diacon, A. H., Tang, S. J., Conradie, F., Danilovits, M., Chuchottaworn,
C., Vasilyeva, I., Andries, K., Bakare, N., De Marez, T., Haxaire-Theeuwes, M.,
Lounis, N., Meyvisch, P., Van Baelen, B., van Heeswijk, R. P., Dannemann, B., &
TMC207-C209 Study Group (2016). Bedaquiline in the treatment of multidrug- and
extensively drug-resistant tuberculosis. *The European respiratory journal*, 47(2), 564–
574. <https://doi.org/10.1183/13993003.00724-2015>
- 6. Xia X. (2023). Horizontal Gene Transfer and Drug Resistance
Involving *Mycobacterium tuberculosis*. *Antibiotics*, 12(9), 1367.
<https://doi.org/10.3390/antibiotics12091367>

- 7. Levillain, F., Poquet, Y., Mallet, L., Mazères, S., Marceau, M., Brosch, R., Bange, F.
C., Supply, P., Magalon, A., & Neyrolles, O. (2017). Horizontal acquisition of a
hypoxia-responsive molybdenum cofactor biosynthesis pathway contributed to
*Mycobacterium tuberculosis* pathoadaptation. *PLoS pathogens*, *13*(11), e1006752.
<https://doi.org/10.1371/journal.ppat.1006752>
- 8. Madacki, J., Orgeur, M., Mas Fiol, G., Frigui, W., Ma, L., & Brosch, R. (2021). ESX-
1-Independent Horizontal Gene Transfer by *Mycobacterium tuberculosis* Complex
Strains. *mBio*, *12*(3), e00965-21. <https://doi.org/10.1128/mBio.00965-21>
- 9. Chiner-Oms, Á., López, M. G., Moreno-Molina, M., Furió, V., & Comas, I. (2022).
Gene evolutionary trajectories in *Mycobacterium tuberculosis* reveal temporal signs of
selection. *PNAS*, *119*(17), e2113600119. <https://doi.org/10.1073/pnas.2113600119>
- 10. Hatfull G. F. (2018). Mycobacteriophages. *Microbiology spectrum*, *6*(5),
10.1128/microbiolspec.GPP3-0026-2018. <https://doi.org/10.1128/microbiolspec.GPP3-0026-2018>.
- 11. Gonzalo-Asensio, J., Pérez, I., Aguiló, N., Uranga, S., Picó, A., Lampreave, C.,
Cebollada, A., Otal, I., Samper, S., & Martín, C. (2018). New insights into the
transposition mechanisms of IS6110 and its dynamic distribution between
*Mycobacterium tuberculosis* Complex lineages. *PLoS genetics*, *14*(4), e1007282.
<https://doi.org/10.1371/journal.pgen.1007282>
- 12. Kremer, K., Glynn, J. R., Lillebaek, T., Niemann, S., Kurepina, N. E., Kreiswirth, B.
392 N., Bifani, P. J., & van Soolingen, D. (2004). Definition of the Beijing/W lineage of
393 *Mycobacterium tuberculosis* on the basis of genetic markers. *Journal of clinical*
*microbiology*, *42*(9), 4040–4049. <https://doi.org/10.1128/JCM.42.9.4040-4049.2004>
- 13. Coll, F., Phelan, J., Hill-Cawthorne, G. A., Nair, M. B., Mallard, K., Ali, S., Abdallah,
396 A. M., Alghamdi, S., Alsomali, M., Ahmed, A. O., Portelli, S., Oppong, Y., Alves, A.,
Bessa, T. B., Campino, S., Caws, M., Chatterjee, A., Crampin, A. C., Dheda, K.,
Furnham, N., ... Clark, T. G. (2018). Genome-wide analysis of multi- and extensively
drug-resistant *Mycobacterium tuberculosis*. *Nature genetics*, *50*(2), 307–316.
<https://doi.org/10.1038/s41588-017-0029-0>
- 14. Zakham, F., Sironen, T., Vapalahti, O., & Kant, R. (2021). Pan and Core Genome
Analysis of 183 *Mycobacterium tuberculosis* Strains Revealed a High Inter-Species
Diversity among the Human Adapted Strains. *Antibiotics (Basel, Switzerland)*, *10*(5),
500. <https://doi.org/10.3390/antibiotics10050500>

- 15. Negrete-Paz, A. M., Vázquez-Marrufo, G., Gutiérrez-Moraga, A., & Vázquez-
Garcidueñas, M. S. (2023). Pangenome Reconstruction of *Mycobacterium*
*tuberculosis* as a Guide to Reveal Genomic Features Associated with Strain Clinical
Phenotype. *Microorganisms*, *11*(6), 1495.
<https://doi.org/10.3390/microorganisms11061495>
- 16. Hurtado-Páez, U., Álvarez Zuluaga, N., Arango Isaza, R. E., Contreras-Moreira, B.,
Rouzaud, F., & Robledo, J. (2023). Pan-genome association study of *Mycobacterium*
*tuberculosis* lineage-4 revealed specific genes related to the high and low prevalence
of the disease in patients from the North-Eastern area of Medellín,
Colombia. *Frontiers in microbiology*, *13*, 1076797.
<https://doi.org/10.3389/fmicb.2022.1076797>
- 17. Kavvas, E. S., Catoi, E., Mih, N., Yurkovich, J. T., Seif, Y., Dillon, N., Heckmann, D.,
Anand, A., Yang, L., Nizet, V., Monk, J. M., & Palsson, B. O. (2018). Machine
learning and structural analysis of *Mycobacterium tuberculosis* pan-genome identifies
genetic signatures of antibiotic resistance. *Nature communications*, *9*(1), 4306.
<https://doi.org/10.1038/s41467-018-06634-y>
- 18. Kayani, M. R., Zheng, Y. C., Xie, F. C., Kang, K., Li, H. Y., & Zhao, H. T. (2018).
Genome Sequences and Comparative Analysis of Two Extended-Spectrum
Extensively-Drug Resistant *Mycobacterium tuberculosis* Strains. *Frontiers in*
*pharmacology*, *9*, 1492. <https://doi.org/10.3389/fphar.2018.01492>
- 19. Advani, J., Verma, R., Chatterjee, O., Pachouri, P. K., Upadhyay, P., Singh, R., Yadav,
426 J., Naaz, F., Ravikumar, R., Buggi, S., Suar, M., Gupta, U. D., Pandey, A., Chauhan, D.
S., Tripathy, S. P., Gowda, H., & Prasad, T. S. K. (2019). Whole Genome Sequencing
of *Mycobacterium tuberculosis* Clinical Isolates From India Reveals Genetic
Heterogeneity and Region-Specific Variations That Might Affect Drug
Susceptibility. *Frontiers in microbiology*, *10*, 309.
<https://doi.org/10.3389/fmicb.2019.00309>
- 20. Chizimu, J. Y., Solo, E. S., Bwalya, P., Tanomsridachchai, W., Chambaro, H., Shawa,
433 M., Kapalamula, T. F., Lungu, P., Fukushima, Y., Mukonka, V., Thapa, J., Nakajima,
C., & Suzuki, Y. (2021). Whole-Genome Sequencing Reveals Recent Transmission of
Multidrug-Resistant *Mycobacterium tuberculosis* CAS1-Kili Strains in Lusaka,
Zambia. *Antibiotics*, *11*(1), 29. <https://doi.org/10.3390/antibiotics11010029>
- 21. Dreyer, V., Mandal, A., Dev, P., Merker, M., Barilar, I., Utpatel, C., Nilgiriwala, K.,
Rodrigues, C., Crook, D. W., CRyPTIC Consortium, Rasigade, J. P., Wirth, T., Mistry,

- 439 N., & Niemann, S. (2022). High fluoroquinolone resistance proportions among
440 multidrug-resistant tuberculosis driven by dominant L2 Mycobacterium tuberculosis
clones in the Mumbai Metropolitan Region. *Genome medicine*, *14*(1), 95.
<https://doi.org/10.1186/s13073-022-01076-0>
- 22. Jabbar, A., Phelan, J. E., de Sessions, P. F., Khan, T. A., Rahman, H., Khan, S. N.,
Cantillon, D. M., Wildner, L. M., Ali, S., Campino, S., Waddell, S. J., & Clark, T. G.
(2019). Whole genome sequencing of drug resistant Mycobacterium tuberculosis
isolates from a high burden tuberculosis region of North West Pakistan. *Scientific*
*reports*, *9*(1), 14996. <https://doi.org/10.1038/s41598-019-51562-6>
- 23. Xiao, Y. X., Liu, K. H., Lin, W. H., Chan, T. H., & Jou, R. (2023). Whole-genome
sequencing-based analyses of drug-resistant Mycobacterium tuberculosis from
Taiwan. *Scientific reports*, *13*(1), 2540. <https://doi.org/10.1038/s41598-023-29652-3>
- 24. Li, D., Liu, C. M., Luo, R., Sadakane, K., & Lam, T. W. (2015). MEGAHIT: an ultra-
fast single-node solution for large and complex metagenomics assembly via succinct
de Bruijn graph. *Bioinformatics*, *31*(10), 1674–1676.
<https://doi.org/10.1093/bioinformatics/btv033>
- 25. Alonge, M., Lebeigle, L., Kirsche, M., Jenike, K., Ou, S., Aganezov, S., Wang, X.,
Lippman, Z. B., Schatz, M. C., & Soyk, S. (2022). Automated assembly scaffolding
using RagTag elevates a new tomato system for high-throughput genome
editing. *Genome biology*, *23*(1), 258. <https://doi.org/10.1186/s13059-022-02823-7>
- 26. Wood, D. E., & Salzberg, S. L. (2014). Kraken: ultrafast metagenomic sequence
classification using exact alignments. *Genome biology*, *15*(3), R46.
<https://doi.org/10.1186/gb-2014-15-3-r46>
- 27. Gurevich, A., Saveliev, V., Vyahhi, N., & Tesler, G. (2013). QUAST: quality
assessment tool for genome assemblies. *Bioinformatics*, *29*(8), 1072–1075.
<https://doi.org/10.1093/bioinformatics/btt086>
- 28. Phelan, J., O'Sullivan, D. M., Machado, D., Ramos, J., Whale, A. S., O'Grady, J.,
Dheda, K., Campino, S., McNerney, R., Viveiros, M., Huggett, J. F., & Clark, T. G.
(2016). The variability and reproducibility of whole genome sequencing technology
for detecting resistance to anti-tuberculous drugs. *Genome medicine*, *8*(1), 132.
<https://doi.org/10.1186/s13073-016-0385-x>
- 29. Phelan, J. E., O'Sullivan, D. M., Machado, D., Ramos, J., Oppong, Y. E. A., Campino,
S., O'Grady, J., McNerney, R., Hibberd, M. L., Viveiros, M., Huggett, J. F., & Clark, T.
G. (2019). Integrating informatics tools and portable sequencing technology for rapid

- detection of resistance to anti-tuberculous drugs. *Genome medicine*, 11(1), 41.
<https://doi.org/10.1186/s13073-019-0650-x>
- 30. Seemann T. (2014). Prokka: rapid prokaryotic genome
annotation. *Bioinformatics*, 30(14), 2068–2069.
<https://doi.org/10.1093/bioinformatics/btu153>
- 31. Tonkin-Hill, G., MacAlasdair, N., Ruis, C., Weimann, A., Horesh, G., Lees, J. A.,
Gladstone, R. A., Lo, S., Beaudoin, C., Floto, R. A., Frost, S. D. W., Corander, J.,
Bentley, S. D., & Parkhill, J. (2020). Producing polished prokaryotic pangenomes with
the Panaroo pipeline. *Genome biology*, 21(1), 180. [https://doi.org/10.1186/s13059-020-](https://doi.org/10.1186/s13059-020-02090-4)
[02090-4](https://doi.org/10.1186/s13059-020-02090-4)
- 32. Brynildsrud, O., Bohlin, J., Scheffer, L., & Eldholm, V. (2016). Rapid scoring of genes
in microbial pan-genome-wide association studies with Scoary. *Genome*
*biology*, 17(1), 238. <https://doi.org/10.1186/s13059-016-1108-8>
- 33. Sandgren, A., Strong, M., Muthukrishnan, P., Weiner, B. K., Church, G. M., & Murray,
487 M. B. (2009). Tuberculosis drug resistance mutation database. *PLoS medicine*, 6(2),
e2. <https://doi.org/10.1371/journal.pmed.1000002>
- 34. WHO. (2021). Catalogue of Mutations in Mycobacterium Tuberculosis Complex and
Their Association with Drug Resistance.
- 35. Feuerriegel, S., Schleusener, V., Beckert, P., Kohl, T. A., Miotto, P., Cirillo, D. M.,
Cabibbe, A. M., Niemann, S., & Fellenberg, K. (2015). PhyResSE: a Web Tool
Delineating Mycobacterium tuberculosis Antibiotic Resistance and Lineage from
Whole-Genome Sequencing Data. *Journal of clinical microbiology*, 53(6), 1908–1914.
<https://doi.org/10.1128/JCM.00025-15>
- 36. Desikan, S., & Narayanan, S. (2015). Genetic markers, genotyping methods & next
generation sequencing in Mycobacterium tuberculosis. *The Indian journal of medical*
*research*, 141(6), 761–774. <https://doi.org/10.4103/0971-5916.160695>
- 37. Brosch, R., Pym, A. S., Gordon, S. V., & Cole, S. T. (2001). The evolution of
mycobacterial pathogenicity: clues from comparative genomics. *Trends in*
*microbiology*, 9(9), 452–458. [https://doi.org/10.1016/s0966-842x\(01\)02131-x](https://doi.org/10.1016/s0966-842x(01)02131-x)
- 38. Mathema, B., Kurepina, N. E., Bifani, P. J., & Kreiswirth, B. N. (2006). Molecular
epidemiology of tuberculosis: current insights. *Clinical microbiology reviews*, 19(4),
658–685. <https://doi.org/10.1128/CMR.00061-05>
- 39. Shitikov, E. A., Bespyatykh, J. A., Ischenko, D. S., Alexeev, D. G., Karpova, I. Y.,
Kostryukova, E. S., Isaeva, Y. D., Nosova, E. Y., Mokrousov, I. V., Vyazovaya, A. A.,

- Narvskaya, O. V., Vishnevsky, B. I., Otten, T. F., Zhuravlev, V. I., Yablonsky, P. K.,
Ilina, E. N., & Govorun, V. M. (2014). Unusual large-scale chromosomal
rearrangements in Mycobacterium tuberculosis Beijing B0/W148 cluster isolates. *PLoS*
*one*, 9(1), e84971. <https://doi.org/10.1371/journal.pone.0084971>
- 40. Flint, J. L., Kowalski, J. C., Karnati, P. K., & Derbyshire, K. M. (2004). The RD1
virulence locus of Mycobacterium tuberculosis regulates DNA transfer in
Mycobacterium smegmatis. *Proceedings of the National Academy of Sciences of the*
*United States of America*, 101(34), 12598–12603.
<https://doi.org/10.1073/pnas.0404892101>
- 41. He, C., Cheng, X., Kaisaier, A., Wan, J., Luo, S., Ren, J., Sha, Y., Peng, H., Zhen, Y.,
Liu, W., Zhang, S., Xu, J., & Xu, A. (2022). Effects of Mycobacterium
tuberculosis lineages and regions of difference (RD) virulence gene variation on
tuberculosis recurrence. *Annals of translational medicine*, 10(2), 49.
<https://doi.org/10.21037/atm-21-686342>. Coscolla, M. & Gagneux, S.
Consequences of genomic diversity in Mycobacterium tuberculosis. *Semin Immunol*
26, 431–444 (2014).
- 42. Coscolla, M., & Gagneux, S. (2014). Consequences of genomic diversity in
Mycobacterium tuberculosis. *Seminars in immunology*, 26(6), 431–444.
<https://doi.org/10.1016/j.smim.2014.09.012>
- 43. Comas, I., Coscolla, M., Luo, T., Borrell, S., Holt, K. E., Kato-Maeda, M., Parkhill, J.,
Malla, B., Berg, S., Thwaites, G., Yeboah-Manu, D., Bothamley, G., Mei, J., Wei, L.,
Bentley, S., Harris, S. R., Niemann, S., Diel, R., Aseffa, A., Gao, Q., ... Gagneux, S.
(2013). Out-of-Africa migration and Neolithic coexpansion of Mycobacterium
tuberculosis with modern humans. *Nature genetics*, 45(10), 1176–1182.
<https://doi.org/10.1038/ng.2744>
- 44. Merker, M., Blin, C., Mona, S., Duforet-Frebourg, N., Lecher, S., Willery, E., Blum,
533 M. G., Rüscher-Gerdes, S., Mokrousov, I., Aleksic, E., Allix-Béguec, C., Antierens, A.,
Augustynowicz-Kopeć, E., Ballif, M., Barletta, F., Beck, H. P., Barry, C. E., 3rd,
Bonnet, M., Borroni, E., Campos-Herrero, I., ... Wirth, T. (2015). Evolutionary history
and global spread of the Mycobacterium tuberculosis Beijing lineage. *Nature*
*genetics*, 47(3), 242–249. <https://doi.org/10.1038/ng.3195>
- 45. Casali, N., Nikolayevskyy, V., Balabanova, Y., Harris, S. R., Ignatyeva, O.,
Kontsevaya, I., Corander, J., Bryant, J., Parkhill, J., Nejentsev, S., Horstmann, R. D.,
Brown, T., & Drobniowski, F. (2014). Evolution and transmission of drug-resistant

- tuberculosis in a Russian population. *Nature genetics*, 46(3), 279–286.
<https://doi.org/10.1038/ng.2878>
- 46. Cepas, V., & Soto, S. M. (2020). Relationship between Virulence and Resistance
among Gram-Negative Bacteria. *Antibiotics*, 9(10), 719.
<https://doi.org/10.3390/antibiotics910071947>.
- 47. Sala, A., Bordes, P. & Genevoux, P. Multiple Toxin-Antitoxin Systems in
*Mycobacterium tuberculosis*. *Toxins*, 6, 1002 (2014).
- 48. Engohang-Ndong, J., Baillat, D., Aumercier, M., Bellefontaine, F., Besra, G. S., Locht,
C., & Baulard, A. R. (2004). EthR, a repressor of the TetR/CamR family implicated in
ethionamide resistance in mycobacteria, octamerizes cooperatively on its
operator. *Molecular microbiology*, 51(1), 175–188. <https://doi.org/10.1046/j.1365-2958.2003.03809.x>
- 49. Zhang, H., Li, D., Zhao, L., Fleming, J., Lin, N., Wang, T., Liu, Z., Li, C., Galwey, N.,
Deng, J., Zhou, Y., Zhu, Y., Gao, Y., Wang, T., Wang, S., Huang, Y., Wang, M., Zhong,
Q., Zhou, L., Chen, T., ... Bi, L. (2013). Genome sequencing of 161 *Mycobacterium*
*tuberculosis* isolates from China identifies genes and intergenic regions associated with
drug resistance. *Nature genetics*, 45(10), 1255–1260. <https://doi.org/10.1038/ng.2735>
- 50. Ramos, J. L., Martínez-Bueno, M., Molina-Henares, A. J., Terán, W., Watanabe, K.,
Zhang, X., Gallegos, M. T., Brennan, R., & Tobes, R. (2005). The TetR family of
transcriptional repressors. *Microbiology and molecular biology reviews* :
*MMBR*, 69(2), 326–356. <https://doi.org/10.1128/MMBR.69.2.326-356.2005>
- 51. Bespiatykh, D., Bespyatykh, J., Mokrousov, I., & Shitikov, E. (2021). A
Comprehensive Map of *Mycobacterium tuberculosis* Complex Regions of
Difference. *mSphere*, 6(4), e0053521. <https://doi.org/10.1128/mSphere.00535-21>
- 52. Qin, L., Wang, J., Lu, J., Yang, H., Zheng, R., Liu, Z., Huang, X., Feng, Y., Hu, Z., &
Ge, B. (2019). A deletion in the RD105 region confers resistance to multiple drugs in
*Mycobacterium tuberculosis*. *BMC biology*, 17(1), 7. <https://doi.org/10.1186/s12915-019-0628-6>
- 53. Wong, S. Y., Lee, J. S., Kwak, H. K., Via, L. E., Boshoff, H. I., & Barry, C. E., 3rd
(2011). Mutations in *gidB* confer low-level streptomycin resistance in *Mycobacterium*
*tuberculosis*. *Antimicrobial agents and chemotherapy*, 55(6), 2515–2522.
<https://doi.org/10.1128/AAC.01814-10>
- 54. Furió, V., Moreno-Molina, M., Chiner-Oms, Á., Villamayor, L. M., Torres-Puente, M.,
& Comas, I. (2021). An evolutionary functional genomics approach identifies novel

- candidate regions involved in isoniazid resistance in *Mycobacterium*
tuberculosis. *Communications biology*, 4(1), 1322. [https://doi.org/10.1038/s42003-](https://doi.org/10.1038/s42003-021-02846-z)
021-02846-z
- 55. Shekar, S., Yeo, Z. X., Wong, J. C., Chan, M. K., Ong, D. C., Tongyoo, P., Wong, S. Y.,
& Lee, A. S. (2014). Detecting novel genetic variants associated with isoniazid-
resistant *Mycobacterium tuberculosis*. *PloS one*, 9(7), e102383.
<https://doi.org/10.1371/journal.pone.0102383>
- 56. Saiboonjan, B., Roytrakul, S., Sangka, A., Lulitanond, V., Faksri, K., & Namwat, W.
(2021). Proteomic analysis of drug-susceptible and multidrug-resistant nonreplicating
Beijing strains of *Mycobacterium tuberculosis* cultured *in vitro*. *Biochemistry and*
*biophysics reports*, 26, 100960. <https://doi.org/10.1016/j.bbrep.2021.100960>
- 57. Maslov, D. A., Shur, K. V., Bekker, O. B., Zakharevich, N. V., Zaichikova, M. V.,
Klimina, K. M., Smirnova, T. G., Zhang, Y., Chernousova, L. N., & Danilenko, V. N.
(2015). Draft Genome Sequences of Two Pyrazinamide-Resistant Clinical Isolates,
*Mycobacterium tuberculosis* 13-4152 and 13-2459. *Genome announcements*, 3(4),
e00758-15. <https://doi.org/10.1128/genomeA.00758-15>
- 58. The CRyPTIC Consortium (2022). Genome-wide association studies of global
*Mycobacterium tuberculosis* resistance to 13 antimicrobials in 10,228 genomes
identify new resistance mechanisms. *PLoS biology*, 20(8), e3001755.
<https://doi.org/10.1371/journal.pbio.3001755>
- 59. Freidlin, P. J., Nissan, I., Luria, A., Goldblatt, D., Schaffer, L., Kaidar-Shwartz, H.,
Chemtob, D., Dveyrin, Z., Head, S. R., & Rorman, E. (2017). Structure and variation
of CRISPR and CRISPR-flanking regions in deleted-direct repeat region
*Mycobacterium tuberculosis* complex strains. *BMC genomics*, 18(1), 168.
<https://doi.org/10.1186/s12864-017-3560-6>
- 60. Huang, Q., Luo, H., Liu, M., Zeng, J., Abdalla, A. E., Duan, X., Li, Q., & Xie, J.
(2016). The effect of *Mycobacterium tuberculosis* CRISPR-associated Cas2 (Rv2816c)
on stress response genes expression, morphology and macrophage survival of
*Mycobacterium smegmatis*. *Infection, genetics and evolution : journal of molecular*
*epidemiology and evolutionary genetics in infectious diseases*, 40, 295–301.
<https://doi.org/10.1016/j.meegid.2015.10.019>
- 61. Wei, J., Lu, N., Li, Z., Wu, X., Jiang, T., Xu, L., Yang, C., & Guo, S. (2019).
The *Mycobacterium tuberculosis* CRISPR-Associated Cas1 Involves Persistence and

Tolerance to Anti-Tubercular Drugs. *BioMed research international*, 2019, 7861695.
<https://doi.org/10.1155/2019/7861695>

62. Yang, F., Xu, L., Liang, L., Liang, W., Li, J., Lin, D., Dai, M., Zhou, D., Li, Y., Chen,
Y., Zhao, H., Tian, G. B., & Feng, S. (2021). The Involvement of *Mycobacterium* Type
III-A CRISPR-Cas System in Oxidative Stress. *Frontiers in microbiology*, 12, 774492.
<https://doi.org/10.3389/fmicb.2021.774492>

63. Wang, G., Song, G., & Xu, Y. (2020). Association of CRISPR/Cas System with the
Drug Resistance in *Klebsiella pneumoniae*. *Infection and drug resistance*, 13, 1929–
1935. <https://doi.org/10.2147/IDR.S253380>

64. Ren, L., Deng, L. H., Zhang, R. P., Wang, C. D., Li, D. S., Xi, L. X., Chen, Z. R.,
Yang, R., Huang, J., Zeng, Y. R., Wu, H. L., Cao, S. J., Wu, R., Huang, Y., & Yan, Q.
G. (2017). Relationship between drug resistance and the clustered, regularly
interspaced, short, palindromic repeat-associated protein genes cas1 and cas2 in
*Shigella* from giant panda dung. *Medicine*, 96(7), e5922.
<https://doi.org/10.1097/MD.0000000000005922>

65. Singh, A., Gaur, M., Sharma, V., Khanna, P., Bothra, A., Bhaduri, A., Mondal, A. K.,
Dash, D., Singh, Y., & Misra, R. (2021). Comparative Genomic Analysis
of *Mycobacteriaceae* Reveals Horizontal Gene Transfer-Mediated Evolution of the
CRISPR-Cas System in the *Mycobacterium tuberculosis* Complex. *mSystems*, 6(1),
e00934-20. <https://doi.org/10.1128/mSystems.00934-20>

A.

B.

C.

Country	Number of Samples
India	2045
Zambia	68
Pakistan	62
China	9

D.

E.

	Lineage 1	Lineage 2	Lineage 3	Lineage 4
Sensitive	232	33	389	143
RR-TB	5	6	11	3
HR-TB	22	1	55	14
MDR-TB	47	121	103	55
Other	29	5	45	15
Pre-XDR-TB	20	591	119	84
XDR-TB	0	25	6	4

**Figure 1: Whole Genome Sequencing (WGS) data filtering and population structure of**

**clinical *Mycobacterial tuberculosis* isolates included in the study. A: Workflow employed**

**to analyse publicly available WGS data. The data analysis consisted of data acquisition,**

**assembling, DR profiling, lineage determination, metagenome detection, scaffolding, de novo**

**assembly QC, functional annotation, and removal of duplicate GFF files. 2601 (m) samples**

**were downloaded for data analysis. Sample in the sets k_{1-4} successfully underwent pre-**

**processing, and showed high quality on various metrics. Upon Venn analysis, 2184 samples**

**were found common in k_{1-4} and were subsequently subjected to gene repertoire analysis and**

**Pan-GWAS (B). The high-quality sample set (n=2184) consisted of Mtb isolates from 4 TB-**

**endemic countries (India, Pakistan, Zambia and China) (C) with varying drug resistance**

**profiles and lineages (D and E). Abbreviations: DR: Drug Resistant; m: the total number of**

[revised manuscript text omitted]

We are thankful to the esteemed reviewers for their time to review and share critical points to improve the manuscript. We have added new data as suggested by the second reviewer and revised the manuscript as per the suggestions of the first reviewer. We hope the revised manuscript will satisfy the reviewers for a favourable response for publication in the *Microbiology spectrum* journal.

Following are our responses to the suggestions that we received after peer-review of our manuscript titled “Pangenome-wide association study reveals selective absence of CRISPR genes (Rv2816c-19c) in drug-resistant *Mycobacterium tuberculosis*“.

Editor’s suggestion: The authors should also consider depositing the raw sequences in a repository like the NCBI Sequence Read Archive (SRA).

Response: Thank you for this suggestion. As explained in the methodology section, the Whole Genome Sequencing data used in this study are already available in the NCBI-Sequence Read Archive (SRA) database with the BioProject accession IDs PRJNA879962, PRJEB41116, PRJEB32684, PRJNA379070, and PRJEB29435. Furthermore, the individual SRA run identifiers of the filtered dataset used for pangenome reconstruction and Pan-GWAS (n=2184) are presented in Supplementary Table S1. We have also included the European Nucleotide Archive (ENA) accession IDs of the sample set from the Netherlands in Supplementary Table S2.

Response to the reviewer’s suggestions:

Responses to the suggestions of Reviewer 1:

Suggestion 1: The rationale of the study should be incorporated in the abstract focusing on why is this study important.

Response: Thank you for pointing this out. We have revised the abstract. We hope the revised abstract conveys the rationale effectively.

Kindly see Page No. 1, Lines 29-32.

Suggestion 2: Add more keywords; remember that the selected keywords should accurately reflect the content of your paper and be relevant to the research community interested in your topic.

Response: Thanks for highlighting it. We have revised the keywords: Pangenome, Drug resistance/AMR evolution, Tuberculosis, GWAS, DR-determinants, Gene repertoire.

We hope the revised keywords reflects the work better.

Kindly refer to Page No. 1, Lines 48-49.

Suggestion 3: The samples size considered for the study have great difference. The authors are suggested to consider the same sample size from all the countries to make the study less biased.

Response: This manuscript deals with the representative data available from the TB endemic countries. For information, the unfiltered samples from India/Zambia/Pakistan/China :: 2232/88/80/201 reduced to a set of 2045/68/62/9 high-quality samples for subsequent analysis. As India contributes to the highest number of drug-resistant tuberculosis cases, studies like the present manuscript from the local population are missing and it is a critical contribution to study in this field. Furthermore, we observed a strong corroboration of the identified genes having extreme odds ratio scores with the existing literature on their involvement in drug resistance, which further strengthens the outcomes of this study. We have described the corroboration in the discussion section of the manuscript using the following references.

Supporting literature:

1. *Molecular microbiology*, 2004, 51(1). 175-188.
2. *Nature genetics*. 2013. 45(10). 1255-1260.
3. *MMBR*. 2005. 69(2). 326-356.
4. *BMC bio*. 2019. 17(1). 7.
5. *AAC*. 2011. 55(6). 2515-2522.
6. *Comm bio*. 2011. 4(1). 1322. and others...

Suggestion 4: In the results section (Page 6, lines 184 - 187), the authors talk about the pangenome with core, shell and cloud genes. Can the authors give their percentages for each category for comparison?

Response: We have added the percentage distribution details for core, softcore, shell and cloud genes in the revised manuscript. Kindly refer
Kindly refer to Page No. 4, Lines 183-186.

Suggestion 5: The authors should mention about the Function annotation and Pathway analysis. It is suggested to the authors that gene ontology terms associated with the interactome should be explained in detail. Provide all the relevant data for the enrichment performed.

Response: Thank you for pointing this out. We have enhanced the gene ontology figure, added the constituent genes in each GO term in the text, and explained the functions relevant to drug resistance in greater detail.
Kindly refer to Page No. 5, Lines 220-235)

Suggestion 6: In the discussion section Page No. 9; Line No. 284-287 provide the citation for the content. Similarly, provide the citation from the existing literature for the content in Line 288-291.

Response: Thanks for pointing out the missing citations. We have included the in-text citations and supporting references in the revised manuscript.
Kindly refer to Page No. 7, Lines 307-310.

Suggestion 7: The authors are suggested to include conclusion section separately in the manuscript focusing on the outcome of the study with future insights.

Response: We thank the reviewer for highlighting this. We have included the conclusion separately that highlights the outcomes of the manuscript.

Kindly refer to Page No. 8, Lines 367-374.

Suggestion 8: The author should consider redo some of the figures. Many figures show low picture quality or distorted. These figures may need better design and arrangement for comprehensive data presentation.

Response: We are sorry for the low-quality or distorted pictures. Based on the reviewer's kind suggestion, we have corrected and redrawn the manuscript figures with high resolution.

The updated figures have been uploaded as separate files in TIFF format.

Suggestion 9: The authors are suggested use some referencing tool in order to include citation. It should be consistent throughout the manuscript.

Response: We are sorry for the incorrect references. We have refined the references in the revised manuscript.

Suggestion 10: Grammatical, language editing and spelling is highly recommended using professional software and/ or language experts.

Response: Based on the suggestion, we have proofread the manuscript with independent researchers and used professional software like Grammarly to revise the manuscript. We hope the revised manuscript is clearer and more concise.

Response to the suggestion of Reviewer 1:

Reviewer 2: The data can be collected from endemic and non-endemic locations.

Response: Thank you for raising this point. In the revised manuscript, we have replicated the WGS data analysis of clinical Mtb isolates from a non-endemic country (Netherlands n=1130). The absence of CRISPR genes in drug-resistant Mtb clinical isolates was observed and the discordant results were likely due to allopatric evolution.

We would like to emphasize that this study is India-centric, and India contributes to the majority of drug-resistant TB patients worldwide. The second point is that longitudinal studies from different populations of geographically diverse clinical Mtb isolates may show independent evolutionary trajectories owing to the general rule of allopatric evolution.

Re: Spectrum00527-24R1 (Pangenome-wide association study reveals selective absence of CRISPR genes (Rv2816c-19c) in drug-resistant Mycobacterium tuberculosis.)

Dear Dr. Ranjan Kumar Nanda:

Your manuscript has been accepted, and I am forwarding it to the ASM production staff for publication. Your paper will first be checked to make sure all elements meet the technical requirements. ASM staff will contact you if anything needs to be revised before copyediting and production can begin. Otherwise, you will be notified when your proofs are ready to be viewed.

Sincerely,
Felix Toka
Editor
Microbiology Spectrum

Reviewer #1 (Comments for the Author):

The authors have responded to all the comments raised.

Reviewer #4 (Comments for the Author):

Dear author,
The review of the study titled Pangenome-wide association study reveals selective absence of CRISPR genes (Rv2816c-19c) in drug-resistant Mycobacterium tuberculosis has been completed.
The results of the study were found to be very successful. I do not have any additional suggestions.
I wish good work.